# Diverging effects of host density and richness across biological scales drive diversity-disease outcomes

Pieter T. J. Johnson [1] ✉, Tara E. Stewart Merrill[1,2], Andrew D. Dean [3] & Andy Fenton [3]

Understanding how biodiversity affects pathogen transmission remains an unresolved question due to the challenges in testing potential mechanisms in natural systems and how these mechanisms vary across biological scales. By quantifying transmission of an entire guild of parasites (larval trematodes) within 902 amphibian host communities, we show that the community-level drivers of infection depend critically on biological scale. At the individual host scale, increases in host richness led to fewer parasites per host for all parasite taxa, with no effect of host or predator densities. At the host community scale, however, the inhibitory effects of richness were counteracted by associated increases in total host density, leading to no overall change in parasite densities. Mechanistically, we find that while average host competence declined with increasing host richness, total community competence remained stable due to additive assembly patterns. These results help reconcile disease-diversity debates by empirically disentangling the roles of alternative ecological drivers of parasite transmission and how such effects depend on biological scale.

Global biodiversity losses have galvanized efforts to understand how changes in communities affect ecological processes, including pathogen transmission (the movement of parasites between hosts; see "transmission" in Box 1). Evidence from theoretical models, empirical surveys, and experimental manipulations indicates that biodiversity shifts can influence transmission through multiple mechanisms relating to the presence, abundance, and identity of alternate hosts, predators, and coinfecting symbionts[1–4]. While initial debates on diversity–disease relationships centered around whether biodiversity generally protects against disease (via the "dilution effect"[5]), more recent syntheses have established that while biodiversity often inhibits parasite transmission, there is considerable variability in the occurrence and magnitude of this effect among systems[6–11]. The challenge now is to understand the drivers of such heterogeneity across systems and delineate under what host, parasite, and environmental conditions changes in diversity amplify or dilute disease risk[12,13].

Developing a predictive framework linking biodiversity and disease requires a mechanistic approach rooted in community ecology that can identify how changes in host functional traits along richness gradients alter transmission in complex natural systems. Two obstacles continue to limit progress toward this objective[13]. First, unlike for many experimental studies, diversity gradients in real-world ecosystems are non-random and multifactorial. Thus, in addition to changes in host species richness, diversity gradients involve concurrent shifts in community composition, host density (or biomass), and the density of non-host taxa (e.g., predators that consume infective stages)—all of which can shape parasite transmission, potentially in counteracting ways (Fig. 1; refs. 14–16). In particular, dilution effects may be expected to occur more often when community assembly is substitutive as opposed to additive[2,11] (Box 1). In the former case, individuals of added species replace those already present, and total host density remains constant. If newly added individuals are of lower average competence

[1]Ecology and Evolutionary Biology, University of Colorado, Boulder, CO, USA. [2]Coastal and Marine Laboratory, Florida State University, St. Teresa, FL, USA. [3]Institute of Infection, Veterinary & Ecological Sciences, University of Liverpool, Liverpool, UK. ✉e-mail: pieter.johnson@colorado.edu

## BOX 1

# Glossary

| Term | General definition | Specific formulation in the study |
|---|---|---|
| **Infection pressure** | Density of infective stages in an environment | Cercaria density, estimated as the product of infected snail density, mean snail size, and size-scaled cercaria yield |
| **Transmission** | Movement of infective stages from one host (snails) to another (amphibians) | Slope of the line relating **infection pressure** (estimated cercaria density) to **infection success** (metacercariae) |
| **Infection success** | Infection load following **transmission** (conditional on **infection pressure**) | Quantified at the individual scale (metacercariae per host) or community scale (metacercaria density summed over hosts) |
| **Disease** | Pathology (symptoms) resulting from infection | Disease links to infection via intensity-dependent pathology; each metacercia increases likelihood of pathology |
| **Biological scale** | Level of biological organization (i.e., spanning molecules to ecosystems) | We focus on the individual scale ('**host perspective**') and the community scale ('**parasite perspective**') |
| **Host perspective** | **Biological scale** representing **infection success** for an individual host | Considers a host's risk of disease by assessing average metacercariae per host (conditional on **infection pressure**) |
| **Parasite perspective** | **Biological scale** representing **infection success** across the host community | Considers a parasite's fitness by assessing total density of metacercariae (conditional on **infection pressure**) |
| **Host richness** | The number of potential host species within a community | Count of amphibian species present in a community |
| **Density** | The density of a species (number of individuals per unit area) | We consider the density of focal hosts (chorus frogs), predators (damselfly larvae), and all amphibians |
| **Competence** | The ability of a host species to support an infection, given exposure | From Stewart Merrill et al. 2022 and scaled between 0 and 1 |
| **Average competence** | The average **competence** of all potential host species in a community | Mean **competence** for all non-endangered amphibians present in a community |
| **Community competence** | The ability of a community to support infection, given exposure | The total **density** of hosts available to a parasite, adjusted by each host species' **competence** |
| **Additive/substitutive community assembly** | Whether total host **density** increases (additive) or remains constant (substitutive) with **host richness** | Regression relating **host richness** (count of amphibian host species present in a community) to total host **density** (the sum of all larval amphibian densities) |

than the individuals they replace, overall transmission is likely to decrease via dilution effects. When assembly is additive, however, increases in richness covary positively with increased host densities because individuals of new species are added to those of the existing community; hence, overall transmission may increase with host richness[15].

Second, the effects of diversity on parasite transmission can have opposing effects depending on biological scale[17]: how community composition influences infection of individual hosts may differ from how it affects total infection success for the parasite across the host community (Fig. 1). Given additive community assembly, an individual host may have a lower per-capita risk of infection in high diversity (i.e., high host density) systems due to encounter dilution[5,18,19], where parasite infective stages are shared among an increasing number of hosts. Concurrently, the increases in total host density that emerge from additive assembly can provide a parasite with a higher probability of contacting a host due to mass action[18]. Thus, while increasing host diversity may reduce infections in a particular host species, total infection success across the community may show little change or increase with diversity. As such, studies that adopt different biological scales to quantify diversity–disease relationships may come to

opposing conclusions, purely due to contrasting patterns occurring at each level. Despite the central importance of both scales for understanding disease, few studies have simultaneously contrasted responses between individual hosts and host communities across naturally occurring diversity gradients[15,17].

By quantifying infections for an entire guild of parasites (larval trematodes) among >17,000 amphibian hosts in 902 communities, we tested the influence of three alternative mechanisms—host richness, host density, and the density of predators that consume infective stages—in driving transmission at two "biological scales": average infection success among individual hosts and total infection success across the host community (Box 1 and Fig. 1). We focused specifically on how diversity affects parasite transmission (Box 1), operationally defined here as the slope of the relationship between the number of trematode infective stages (cercariae) and the number of established parasites in amphibian hosts. Dilution effects generally act through changes to the process of transmission, for example, through encounter reduction[5]; thus, although often more difficult to measure in field systems, evaluating changes in transmission offers a more direct parallel to predictions from theory and results from experimental studies. To help address the extent to which transmission

## Infection pressure

**Fig. 1 | Progress in diversity–disease research requires a mechanistic understanding of the factors that affect transmission as well as a consideration of biological scale.** Changes in community diversity can cause shifts in host species richness, host density, and the consumption of infective stages by predators (middle of the figure). By quantifying these diversity-linked mechanisms for 902 communities, we test their importance for parasite transmission from snails to amphibians. Our analyses occur at two biological scales. We quantify the average number of parasites per host to estimate infection success at the individual host scale. This captures the host perspective by addressing how changes in diversity amplify or dilute a host's risk of acquiring infection and experiencing disease. We also quantify the total density of parasites summed across the host community to estimate infection success at the community scale. This captures the parasite perspective by addressing how changes in diversity increase or decrease a parasite's ability to successfully infect a host (influencing the potential of the parasite population to cause disease in the future). Images of adult amphibians were created with BioRender.com.

changes were driven by richness per se or concurrent shifts in community composition, we complemented field surveys with experimentally derived estimates of "competence" (transmission potential; Box 1) for each host–parasite combination, thereby allowing functional estimates of each host species' and host communities' capacity to support transmission. Shifts in average or community competence are hypothesized to be one of the major mechanisms underlying dilution effects[20,21], yet such measurements are often lacking for animal disease systems. Incorporating infection data from multiple parasite species and across replicate communities offers a unique opportunity to evaluate the ecological mechanisms underlying diversity–disease relationships and assess the outcome of such effects for both host disease risk and parasite infection success.

## Results

### Infection success at the individual host scale (the host perspective)

Increasing host richness (Box 1), but not focal host density or the density of predators (Box 1), consistently reduced "infection success" in individual hosts (the "host perspective"; Box 1) for all four parasite species examined. For the most common amphibian host (chorus frog, *Pseudacris regilla*, a key indicator species for infection), infection success (quantified as metacercariae per host) of each trematode species was strongly and positively related to "infection pressure" (Box 1) from snail intermediate hosts (estimated density of infective trematode cercariae in a given pond, quantified from the density of infected snails, average snail size, and size-adjusted number of cercariae released based on snail length-to-cercariae regressions; see *SI: Formulation of predictor variables* and Table S6) (Fig. 2). The coefficients ± 1 SE for infection pressure (mean centered and scaled) on infection success in chorus frogs varied among parasites, with values of 3.31 ± 0.251 (*Alaria marcinae*), 2.247 ± 0.189 (*Cephalogonimus americanus*), 0.964 ± 0.101 (*Echinostoma* spp.), and 2.184 ± 0.166 (*Ribeiroia ondatrae*, all parasites hereafter referred to by their genus name) (see Table S1). However, each parasite's infection success was also moderated by a negative interaction between infection pressure and host richness, indicating an inhibitory effect of richness on parasite transmission. Thus, chorus frog hosts in communities with higher amphibian host richness had fewer metacercariae (lower infection success) than those in species-poor communities, after controlling for infection pressure (Fig. 2; infection pressure x host richness coefficients: $-0.394 \pm 0.188$, $P = 0.036$ [*Alaria*], $-0.374 \pm 0.157$, $P = 0.0174$ [*Cephalogonimus*], $-0.320 \pm 0.100$, $P = 0.0014$ [*Echinostoma*], $-0.711 \pm 0.137$, $P < 0.00001$ [*Ribeiroia*]). The interaction between infection pressure and focal host density was significant only for *Ribeiroia* ($0.251 \pm 0.121$, $P = 0.0376$), while neither the density of predators nor its interaction with infection pressure influenced infection success for any other parasite species (Fig. 2 inlays). Diagnostic analyses of these models showed no evidence of overdispersion, major outliers, or collinearity (all VIFs < 1.5; correlation matrices of predictors available in Table S8). The marginal $R^2$ values for the final models were 0.37 (*Alaria*), 0.26 (*Cephalogonimus*), 0.20 (*Echinostoma*), and 0.29 (*Ribeiroia*) (see Table S1 for full model results).

Extending this analysis to all five non-endangered host species reinforced these findings. Increasing host richness consistently reduced infection success (metacercariae per host) for individuals of all amphibian species via a negative interaction with infection pressure for all four parasites (Fig. 3; infection pressure × host richness $\chi^2$: 4.895, $P = 0.027$ [*Alaria*], 9.52, $P = 0.002$ [*Cephalogonimus*], 4.739, $P = 0.0294$ [*Echinostoma*], 28.45, $P < 0.00001$ [*Ribeiroia*]). For the parasite *Ribeiroia*, there was an additional 3-way interaction between infection pressure, host richness, and amphibian species identity ($\chi^2 = 12.699$, df = 4, $P = 0.0128$), such that increasing host richness had a more protective effect for bullfrogs (*Rana catesbeiana*) relative to other amphibian hosts (Fig. 3). None of the other three parasites had significant three-way interactions, suggesting that increasing host richness generally decreased infection success in individual hosts, regardless of amphibian species identity (see Tables S2 and S3).

### Infection success at the community scale (the parasite perspective)

Total parasite density (representing the "parasite perspective" [Box 1] and quantified as the sum of each host species' average infection load multiplied by its larval density) was positively influenced by total host density and negatively affected by the infection pressure × host richness interaction. As expected based on density-dependent transmission[18], total host density had consistently positive main effects and interactions with infection pressure in predicting total

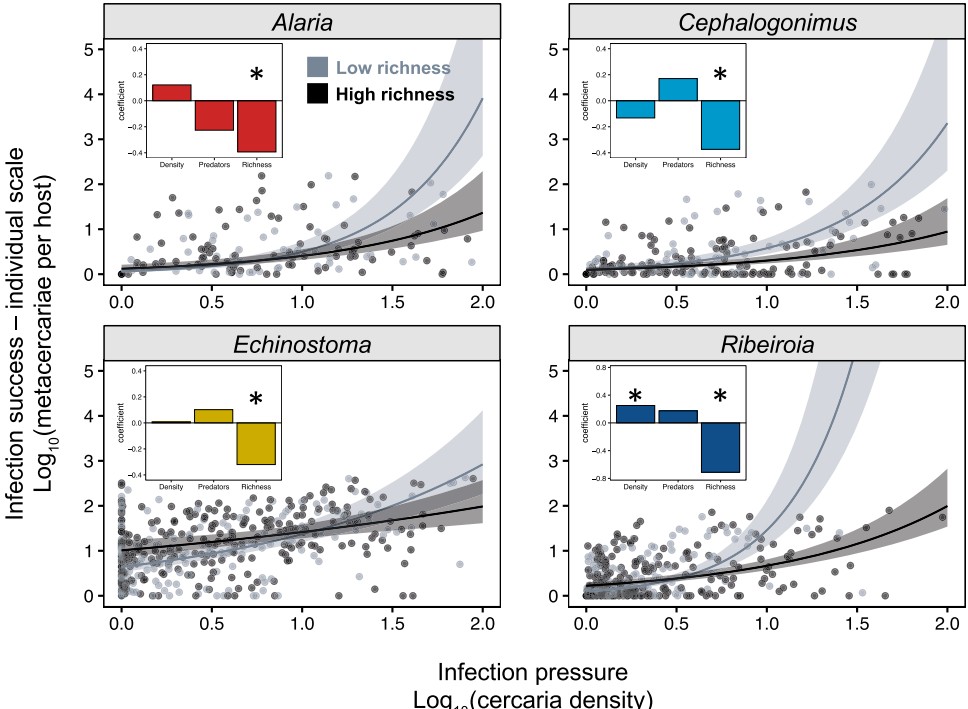

**Fig. 2 | Host richness broadly decreases infection success at the individual host scale for four trematode parasites.** For each trematode species (labels above panels), the average number of parasites (metacercariae) per chorus frog (*Pseudacris regilla*) in a given site and year is positively predicted by infection pressure (a proxy for the density of infective cercariae based on the density, average size, and prevalence of infection in snail intermediate hosts). The slope of this relationship is steeper in low-richness amphibian communities (1 species; gray line) relative to high-richness communities (4 species; black line). Smoothed lines represent marginal effects predicted from ggeffects (i.e., lines denote the effect of host richness on transmission with host and predator densities held at their mean values). Predictor and response variables (infection pressure and infection success, respectively) are $\log_{10} + 1$ transformed. Inlays present coefficients for the interactive effects of host density, predator density, and host richness with infection pressure (asterisks [*] show significant [two-tailed $P$ value < 0.05; unadjusted] interaction terms with infection pressure from generalized linear mixed models). To display all raw datapoints, while maintaining visual alignment with the low and high richness groupings of the fitted lines, raw richness values are binned (gray points are communities of 1 or 2 species, black points are communities of 3 or more species). For all four parasites, the interaction between infection pressure and host richness was negative and significant. Shading represents the standard error of the fit regression. Sample sizes (number of site-year combinations) varied by parasite (*Alaria*: $n = 346$ site-years; *Cephalogonimus*: $n = 495$ site-years; *Echinostoma*: $n = 432$ site-years; *Ribeiroia*: $n = 496$ site-years). Source data are provided as a Source data file.

parasite density within the host community (see Table S4). After accounting for the positive influence of host density, increases in host richness led to progressive declines in parasite infection success across the amphibian community, as supported by the models for *Ribeiroia*, *Cephalogonimus*, and *Alaria* (Table S4; infection pressure × host richness coefficients: $-0.0886 \pm 0.0341$, $P = 0.009$ [*Alaria*], $-0.0715 \pm 0.0238$, $P = 0.0026$ [*Cephalogonimus*], $-0.1082 \pm 0.0282$, $P = 0.00013$ [*Ribeiroia*]). There was no evidence that predator density inhibited infection success for any of the parasite species (see Table S4).

Community competence (Box 1)—which extends beyond host richness by quantitatively combining information on species composition, host density, and experimental measurements of host suitability for supporting infection—provided the best model fit for total parasite density (infection success at the community scale). For three of the four parasite species, the model with community competence had a significant, positive interaction effect with infection pressure and a ΔAIC of between 12 and 28 units lower than the alternative model with richness and density (see Table S5; infection pressure x community competence coefficients: $0.174 \pm 0.0354$, $P < 0.00001$ [*Alaria*], $0.242 \pm 0.022$, $P < 0.00001$ [*Cephalogonimus*], $0.071 \pm 0.039$, $P = 0.0714$ [*Echinostoma*], $0.113 \pm 0.0296$, $P = 0.00014$ [*Ribeiroia*]). These models explained 45 to 65% of the variation in total infection success (marginal $R^2$ values). Whether sites with threatened amphibians were included in the analysis had little effect on the results.

## Contrasting diversity−disease relationships across biological scales

Comparing the predicted effects of host richness on infection success revealed contrasting patterns between biological scales: progressive increases in host richness were associated with lower average infection success in individual hosts (Fig. 4 solid lines) but led to maintenance of (or slight increases in) total parasite infection success at the community scale (Fig. 4 dashed lines). These contrasting effects were driven by the differing impacts of host density at each scale. At the individual host scale, the density of focal hosts had little influence on infection risk after accounting for the negative effect of host richness. At the community level, however, total host density—which increased with host richness following an additive assembly pattern (see Fig. S2)—associated positively with total parasite infection success. Hence, at the community scale, the increases in transmission that resulted from increases in total host density were balanced against any inhibitory effects of host richness (see statistical results in "Infection success at the community scale" and Table S4). Incorporating the experimentally estimated values of competence per species and per community provided additional mechanistic insight on this point. Increases in host richness were associated with decreases in average host competence (i.e., lower host suitability; Fig. 5), suggesting less-competent species tended to be included as communities became richer. However, community competence (which incorporated rather than controlled for host density) remained largely constant as richness increased, reflecting the approximate balancing of host density increases and

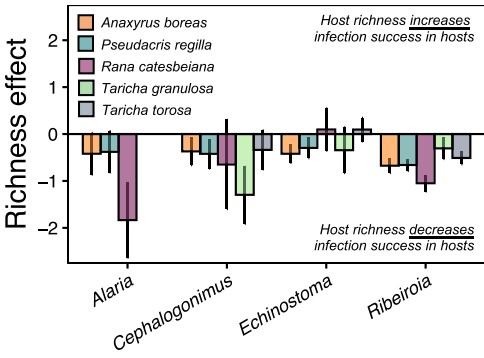

**Fig. 3 | The protective effects of host richness among amphibian species.** Each bar presents the mean coefficient (±1 SE) from a model evaluating the effect of host richness on parasite transmission. Models were run for each of the four trematodes (on the *x*-axis) and in each of the five non-endangered amphibian host species (represented by colors and in the order of the inlaid key). Negative values (bottom region of the plot) indicate that richness inhibits transmission to a given host species (i.e., reduced infection success in individual hosts of that species). Positive values (top region of the plot) indicate that richness increases transmission to a given host species. For the trematode *Ribeiroia*, there was a significant interaction between host richness and host species identity, such that the protective effects of host richness were greater for bullfrogs (*Rana catesbeiana*) and weaker for newt species (*Taricha torosa* and *T. granulosa*). Sample sizes (number of site-year-species combinations) used to generate coefficients varied by parasite (*Alaria*: n = 478; *Cephalogonimus*: n = 1088; *Echinostoma*: n = 1079; *Ribeiroia*: n = 1091). Source data are provided as a Source data file.

average competence decreases (Fig. 5), leading to few effects on overall community-wide transmission success.

## Discussion

Our study advances research on diversity–disease relationships by simultaneously testing the importance of alternative mechanisms linking biodiversity and parasite transmission in a natural system and evaluating how such effects vary with biological scale. By collecting a high-resolution dataset of host and parasite assemblages across 902 communities over 11 years, we assessed the influence of changes in host species richness, host density, and predators of parasite propagules on the transmission of the four most common trematode species in amphibian communities. We analyzed the role of each hypothesis at two distinct biological levels: the individual scale, which offers direct insight into host disease risk, and the community scale, which provides insight into parasite fitness. These analyses revealed that increases in host richness consistently reduced infection success in individual hosts across multiple host and parasite species. At the community scale, however, total parasite infection success was maintained—or increased slightly—as communities became more species rich. This stemmed from the tension between an increase in total host density with higher richness (due to additive community assembly) coupled with the decrease in average host competence. Hence, our results emphasize the concurrent roles of both host species composition and host density in driving responses along diversity gradients. These insights were revealed by measuring density and community effects on the process of transmission (i.e., parasites' ability to move among hosts), rather than on static measures of infection that do not control for exposure.

At the scale of individual hosts, increases in host community richness were associated with a steep reduction in infection success for all parasite species studied. That is, greater host diversity resulted in fewer metacercariae per host than predicted based on infection pressure (i.e., the density of infective cercariae per pond). This effect

was broadly consistent among parasite and host species in the community. In contrast, changes in the density of focal amphibian hosts (chorus frogs) or the abundance of predators (represented here by the density of damselfly larvae) had few detectable effects on infection, despite evidence regarding the potential influence of each pathway in smaller-scale experiments[22,23]. Mechanistically, the protective influence of richness likely stems from encounter reduction, in which alternative (and often less-competent) host species alter the rates of contact between infective parasites and suitable host individuals[5]. It is noteworthy that this diluting effect was evident even though communities assembled additively (i.e., density increased with richness), rather than substitutively (in which added individuals replace existing ones)[2,24]. When considering transmission at the community scale, net parasite infection success also decreased with higher host richness. However, this effect was only apparent while controlling for host density, which covaried positively with richness and itself had a strongly positive effect on total parasite infection success. Higher host densities likely afford increased opportunities for infective cercariae to locate a host before expiring[25,26]. Because total host density (summed among all host species) increased with richness (see Fig. S4), such that richer assemblages supported more host individuals, the inhibitory effects of richness on parasite infection success were broadly offset by the positive changes associated with density. As a result, total parasite infection success was relatively stable (or increased slightly) in richer communities, even while infections in individuals declined (see also refs. 15,27,28). Our use of this metric is novel in diversity–disease theory; while prevalence and mean infection load have been common responses measured at both the host and community scale[17,27], total parasite infection success is rarely invoked as a response to diversity. Yet, taking the parasite perspective (considering how many parasites successfully infect hosts as a consequence of host diversity) has important implications for the recruitment and future transmission of the parasite population[26] and for its potential evolution within the host community. This finding emphasizes that reductions in individual host infections do not necessarily translate into population-level decreases in overall parasite transmission.

The contrasting and scale-dependent nature of diversity effects on infection detected here provide insights that can help resolve ongoing uncertainty over the expected influence of diversity changes on disease. In the current study system, increases in host richness broadly reduced infections in individuals while having few adverse effects on total parasite infection success (and thus parasite fitness) at the community scale. Importantly, however, the generalizability of such effects will depend critically on patterns of community assembly[15]. Theory (and previous empirical data) suggest that the negative effects of species richness on pathogen transmission are most likely to manifest in communities with substitutive (compensatory) assembly, in which individuals of newly added species are associated with decreases in the density of existing species (e.g., through competition or predation)[24]. If added individuals have lower average parasite competence than the individuals they replace, reductions in transmission are expected. Conversely, communities that assemble additively with an increase in total host density with increasing species richness are expected to show either weak or positive relationships between richness and transmission or disease (especially over large spatial or temporal scales)[15]. Here, richer communities exhibited lower average values of host competence for each parasite studied (Fig. 5), which was explained by the progressive addition of less-competent host species as richness increased due to processes such as colonization-defense tradeoffs in hosts or local adaptation by parasites[29]. The degree to which this holds true in other systems will depend on the shape of the relationships between richness and both host density (additive, substitutive, or saturating) and average host competence[2,30]. Thus, if host density remains stable with richness or saturates quickly (e.g., due to limited resource availability), decreases

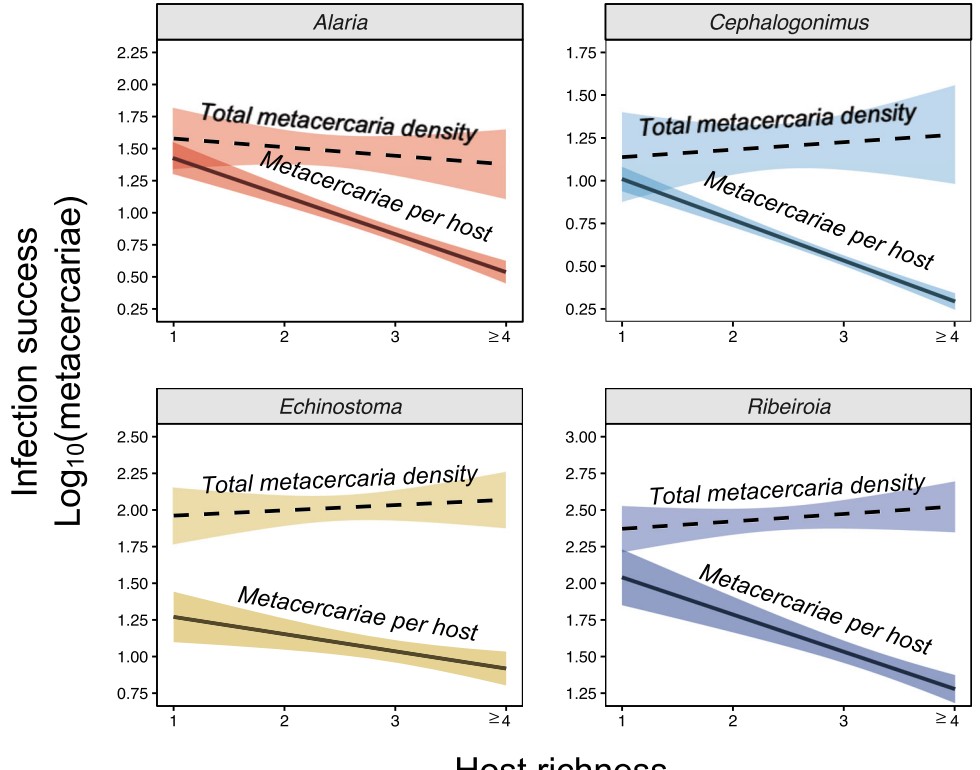

**Fig. 4 | Scale-dependent effects of host richness on parasite transmission.** For each trematode, model-predicted values illustrate how host richness alters infection success in individual hosts (host perspective, solid line) versus communities (parasite perspective, dashed line). While host richness reduced metacercariae per host for each parasite, host richness maintained total metacercariae densities owing to the counteracting effects of host richness on average competence (negative) and total host density (positive). Predictions (central regression lines) were generated for all amphibian hosts at high infection pressure ($\log_{10}$-transformed estimated cercaria density = 2), where shading represents the standard error of the predicted relationship. Sample sizes used to generate the model predictions varied by parasite and biological scale. As in Fig. 3, at the individual host scale, the unit of replication was the site-year-species combination (*Alaria*: $n = 478$; *Cephalogonimus*: $n = 1088$; *Echinostoma*: $n = 1079$; *Ribeiroia*: $n = 1091$). At the community scale, the unit of replication was the site-year combination (*Alaria*: $n = 175$; *Cephalogonimus*: $n = 250$; *Echinostoma*: $n = 259$; *Ribeiroia*: $n = 252$). Source data are provided as a Source data file.

in average host competence will lead more directly to reductions in net parasite transmission, in contrast to the current findings.

An important distinction between the current study and many field-based investigations of diversity–disease relationships is that our analyses focused on how richness altered the process of parasite transmission (here, the capacity of infective parasites to move from snail hosts to amphibians). In contrast, studies that test for correlations between richness and a "snapshot" measure of infection (or disease) offer insight into the outcome of transmission (i.e., the product of infection pressure and infection success). Focusing on the process of transmission, however, provides an additional opportunity to quantify the influence of hypothesized diversity mechanisms in direct parallel to many theoretical and experimental studies, which evaluate how diversity affects infection while controlling for parasite exposure. We emphasize though that our study examined one step in the larger infection process—that between snail and amphibian hosts. Hence, an important future step will involve extending this approach to additional stages of infection with the aim of developing a more comprehensive understanding of the net effects of diversity on disease.

The importance of developing a more mechanistic approach to understanding parasite spread within complex ecological communities has perhaps never been more apparent than today[31,32]. The ongoing introduction of invasive pathogens into new environments, coupled with the spillover of infections into new hosts, illustrate the potentially devastating and seemingly unpredictable consequences for humans and wildlife alike. While debates are often framed around whether biodiversity losses will consistently increase disease risk, such controversies have broadly exposed both the knowledge gaps and

tremendous opportunities for studying disease processes in ecological communities. The present study illustrates the explanatory influence of host competence in understanding shifts in parasite transmission. By integrating information on host competence and host density, "community competence" offered the single-best metric for predicting changes in infection success across diversity gradients and among parasite species. This approach revealed that the effects of richness on parasite infection, both for individual hosts and for entire communities, stemmed primarily from shifts in host density and species composition, each of which changed predictably along host diversity gradients. Whether the same effects are seen in higher richness communities, where stochastic effects may make community assembly much less predictable, is a priority for future investigation. Such findings underscore the importance of more comprehensive investigations into the variables that drive changes in competence as a step toward advancing the community ecology of disease[33].

## Methods
### Compliance
All sampling and experiments conducted as part of this study comply with local, state, and federal regulations. Animal-related research was approved by the University of Colorado Institutional Animal Care and Use Committee (IACUC). Sampling permissions were in compliance with local (California State Parks, East Bay Regional Parks District, East Bay Municipal Utilities District, Santa Clara County Parks, Open Space Authority, Midpeninsula Open Space), state (California Department of Fish and Wildlife), and federal authorities (US Fish and Wildlife Service).

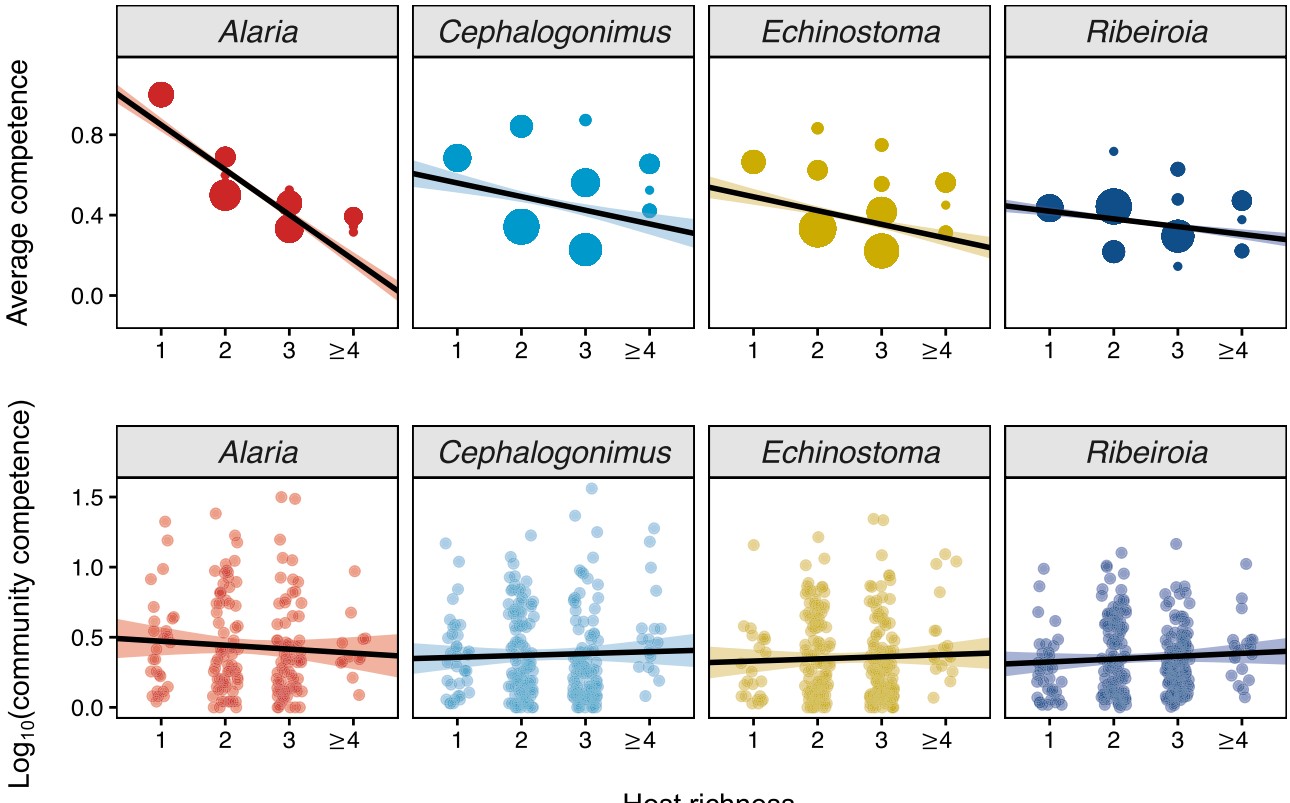

**Fig. 5 | Relationship between amphibian host species richness and average host competence (top row) and community competence (bottom row).** Competence values were derived from experimental exposure trials and scaled between 0 and 1. To calculate the average competence for a given parasite at a given pond, we took the average competence value based on the presence of each species (Eq. 1). Point size reflects the frequency of different community composition configurations. Community competence adjusts the absolute density of the community based on the competence values of the species present (Eq. 2) and, therefore, incorporates information on host identity, competence, and density. Shading around mean regression lines (black) represents the standard error of the fit regression. Sample sizes (number of communities) vary by parasite (*Alaria*: $n = 175$ communities; *Cephalogonimus*: $n = 250$ communities; *Echinostoma*: $n = 259$ communities; *Ribeiroia*: $n = 252$ communities). Source data are provided as a Source data file.

## Study system

Pond ecosystems represent an ideal setting in which to evaluate the mechanistic processes underlying diversity–disease relationships because their small size, well-defined boundaries, and tractable number of taxa facilitate extensive community-level replication. Larval amphibians developing within these systems are infected by a diverse assemblage of parasites[34,35], of which larval trematodes often account for the majority of observed metazoan infections[36–38]. Digenetic trematodes have complex life cycles involving sequential transmission among host species embedded in ecological food webs, including a molluscan first intermediate host (often a snail), a vertebrate or invertebrate second intermediate host, and a vertebrate definitive host[39,40]. Trematode-infected snails release free-swimming cercariae that have <24 h to find a suitable subsequent host, such as a larval amphibian[35,41]. The resultant infection load (i.e., number of parasites per host) determines host pathology and the transmission opportunities to definitive hosts. Certain trematode species, such as *Ribeiroia ondatrae*, can cause substantial mortality and limb malformations among infected amphibian populations[28,42,43].

## Field sampling

Between 2009 and 2019, we sampled 224 ponds distributed across public and private properties in the East Bay region of California, including Contra Costa, Alameda, and Santa Clara counties. We focused on semi-permanent to permanent ponds <3 ha in surface area. Land cover is predominantly annual grasslands and oak woodlands, and most ponds were built or modified to support cattle. Each pond

was sampled twice per year. On the first "initial conditions" visit (May–June), we characterized the aquatic community and quantified the density of larval amphibians, infected snail intermediate hosts, and invertebrate predators (e.g., damselfly larvae; see *SI: Pond sampling*). During the second "transmission assessment" visit (late June and July), we collected 10–15 late-stage amphibian larvae or recently metamorphosed individuals from all non-endangered host species for dissection. The amphibian species included Pacific chorus frogs (*P. regilla*), western toads (*Anaxyrus boreas*), bullfrogs (*Rana catesbeiana*), rough-skinned newts (*Taricha granulosa*), and California newts (*Taricha torosa*). We focused on animals at or nearing metamorphosis to provide a standardized life stage for quantifying water-borne infections acquired during larval development. The density of damselfly larvae was used to represent predation risk on trematodes based on previous experimental studies identifying them as among the most effective consumers of cercariae[22,23,44].

## Parasite quantification

Each collected amphibian host was dissected to identify and quantify larval trematodes (metacercariae and mesocercariae; although, for simplicity, we refer to both as "metacercariae"). We placed particular emphasis on four taxa: *Ribeiroia ondatrae*, *Alaria marcinae*, *Cephalogonimus americanus*, and *Echinostoma* spp. (referred to by their genus names throughout the remainder of the manuscript). These parasites comprise 95% of observed macroparasitic infections in juvenile amphibians from this system[39] and each uses *Helisoma trivolvis* snails as first intermediate hosts (*Echinostoma* can also use *Physa* spp.). To

generate an estimate of infection pressure for developing amphibian larvae, we measured infection prevalence among 50–100 dissected snail intermediate hosts (*H. trivolvis* and *Physa* spp.) over the course of each visit (~50 per visit of each taxon). Snail infections were identified based on characteristics of the specific cercaria morphotype[45] and complementary genetic sampling. Immature (prepatent) infections without identifiable cercariae were not included in estimates of infection prevalence.

## Experimental infections to quantify host competence

We complemented field surveys with controlled laboratory experiments to estimate competence for each of the twenty host-parasite interactions. We collected recently deposited amphibian egg masses or reproductive adults and allowed them to lay eggs in the laboratory. Hatching larvae were maintained in carbon-filtered, UV-sterilized tap-water at 22 °C until being assigned randomly to exposure by one of four trematode taxa (*Ribeiroia*, *Alaria*, *Cephalogonimus*, *Echinostoma*) and one of five ecologically relevant exposure dosages (0 [control], 20, 40, 100, or 200 cercariae). Details of the infection assays are provided by Stewart Merrill and colleagues[46]. In brief, snails (*H. trivolvis*) naturally infected with trematodes were collected from field sites and allowed to release free-swimming cercariae. Harvested cercariae were then counted under a stereodissecting microscope and administered to a 1.5 L container with an individual amphibian larva.

We calculated competence as the product of the dose–response curves for host susceptibility (percentage of administered cercariae that established and persisted as metacercariae) and host survival (likelihood the host survived)[46]. The area under the competence dose-response curve was integrated to generate a standardized estimate of competence for each host-parasite combination. For each amphibian assemblage identified in the initial conditions field surveys, we calculated both the "average competence" of hosts in the community ($\bar{c}$; Box 1) and the "community competence" ($d_c$; Box 1) value as:

$$\bar{c} = \frac{1}{n}\sum_{i=1}^{n} c_i \qquad (1)$$

$$d_c = \sum_{i=1}^{n} c_i d_i \qquad (2)$$

where $c_i$ is the competence of host species $i$, $n$ is the number of species in the community, and $d_i$ is the density of species $i$ (number caught per netsweep). We scaled all $c_i$ prior to inclusion, such that the most competent host for a given parasite had a value of 1. While prior calculations of community competence (i.e., ref. 47) have consisted of a density-weighted average, the calculation in the current study uses absolute density. In this sense, the total density of hosts available to the parasite is adjusted based on the competence of the species present.

## Statistical analysis

We conducted analyses of transmission at two biological scales to capture both the individual host perspective and the parasite perspective (see Box 1 and Fig. 1). First, we evaluated infection success in host individuals, where our response variable was the average number of metacercariae per host. Because infection load per host predicts pathology, this analysis relates directly to disease risk experienced by individuals. Second, we evaluated the infection success of parasites across the complete host community, where our response variable was the total parasite density within a pond (metacercaria densities summed across all co-occurring hosts in a community). This analysis captures total parasite transmission success at the host community scale and is thus most relevant to parasite fitness. Given that parasite fitness shapes parasite population size, this metric carries information about future disease potential. For both biological scales, our focus was on

the capacity of parasite infective stages (cercariae) to move from snail intermediate hosts into suitable amphibian hosts (see "transmission" in Box 1). Thus, all models included a term for infection pressure related to the density of trematode cercariae released from snail intermediate hosts per day. Because this value is difficult to measure directly, we combined information on snail infection prevalence, snail density, snail average size, and the relationship between cercariae release and snail size to create a proxy variable. We calculated the density of infected snails as the product of snail density (average number captured per dipnet sweep) and the prevalence of snails exhibiting infection (number infected divided by number dissected). Recognizing that the number of cercariae emerging from an infected snail can vary substantially as a function of snail size[48], we used species-specific regression equations to link average snail size at a site to the expected number of cercariae released over 24 h (as quantified from a subset of infected snails for each trematode species under standardized conditions, see Table S6). The product of these terms–snail density, infection prevalence, mean size, and the size-specific slope of cercariae release–were used to generate estimates of cercaria density for each trematode species and site-by-year observation (see *SI: Formulation of predictor variables for additional details*). Preliminary investigations indicated that this proxy better fit observed infection data than either infection prevalence in snails or the density of infected snails alone (Table S7). After $\log_{10}$-transformation, the resulting estimate of infection pressure was included in all models as a predictor of amphibian infection at both scales.

We investigated a core set of mechanisms with the potential to reduce transmission between infected snails and individual hosts by lowering host exposure (i.e., factors that prevented cercariae from encountering a host and establishing successfully). We incorporated the following variables based on theory or previous empirical data: host richness (the total number of amphibian species detected as larvae based on all survey methods), host density (average number of focal host larvae or total amphibian larvae captured per netsweep), and the density of aquatic predators (i.e., damselfly larvae) known to consume trematode cercariae[22,23] (Fig. 1). Our analyses sought to evaluate how these community-based mechanisms altered transmission, as assessed by the relationship between infection pressure and infection success, both within individual hosts and across the entire host community. We interpreted a variable (e.g., density, richness, predators) to alter transmission if we detected a significant interaction between that term and infection pressure.

**Infection success at the individual host scale (the host perspective).** We first asked how infection success varied for individual chorus frogs (*P. regilla*), which are the most common amphibian host and represent an indicator species for infection. We ran separate models for each trematode species in turn, for which the response variable was the total number of parasites (metacercariae) among all *P. regilla* in a given pond, incorporating an offset term for the number of dissected hosts to convert this value to "parasites per host" while maintaining the discrete nature of the data. We modeled parasite counts as an over-dispersed Poisson distribution by including an observation-level random effect to account for aggregation. Pond identity and year of sampling were included as random intercept terms to account for sources of autocorrelation.

For the models of each parasite species, we included our three focal community metrics (host richness, host density, and predator density) as predictor variables, with each term included as an interaction with infection pressure; we considered the main effects of each term to be less meaningful than the degree to which they altered the slope of the cercariae-to-amphibian infection relationship (i.e., we expected the intercept to be close to the origin when infection pressure was zero). All numeric predictors were mean-centered and scaled (divided by 1 SD) prior to inclusion, and we ensured that incorporated

terms were not collinear (absolute value of $r < 0.7$; correlation matrices of predictor terms provided in Table S8). Only ponds that supported a given parasite were included in analyses to focus on factors affecting transmission, rather than colonization. All analyses used the R package glmmTMB[49] and statistical tests of significance were always two-tailed. For model assessments, we used the R package performance to calculate the marginal and conditional $R^2$ values, estimate variance inflation factors for predictors, test for overdispersion, and inspect normality of the random effects[50]. Plots of model predictions (Fig. 2) include marginal effects, estimated with the ggmeans package.

We conducted a second analysis at the individual host scale that included all host species in one model, asking whether the influence of richness on per-host infection was consistent among amphibian host species. Here we used the total number of parasites (metacercariae) in each species in a pond as the response variable (with an offset for the number of individuals examined to convert to parasites per host) and included a categorical variable identifying the relevant amphibian species identity. We focused specifically on testing how richness effects differed among host species, and so, in addition to the amphibian species identity term, we incorporated fixed effects for host richness and infection pressure, a pairwise interaction between infection pressure and host richness, and a three-way interaction between infection pressure, host richness, and amphibian species identity. Populations of different host species were nested within ponds using a random intercept term.

**Infection success at the community scale (the parasite perspective).** For analyses focused on the host community scale (Fig. 1), the response variable was the density of established parasites summed across all hosts in the same pond. We again tested how our three community metrics, host richness, total host density, and aquatic predators, interacted with infection pressure to determine the total density of successfully established parasites, quantified as the sum of each host species' infection load (parasites per host, from dissections during the transmission assessment visit) multiplied by its larval density (hosts per netsweep, collected during the initial conditions visit). Because this quantity was continuous rather than discrete, we modeled it as a Gaussian response after $\log_{10}$-transformation (+1). Pond identity and sample year were incorporated as random intercept terms, and we only included site-year combinations in which host species detected in netsweep surveys also had assessments of infection load.

When considering parasite infection success across the host community, interspecific differences in host suitability also matter; transmission is a function not only of whether a parasite can contact a host but also whether it can successfully infect it. We therefore incorporated community-level estimates of community competence, $d_c$, which simultaneously integrate information about host density, species composition, and the functional suitability of hosts for supporting infection (see *Experimental infections to quantify competence*). For each parasite species, we compared the explanatory power of a model containing host density, host richness, and their interactions with infection pressure, with a model including community competence and its interaction with infection pressure. Through this comparison, we aimed to assess the degree to which a model comprising an aggregate measure of overall community competence (integrating host identity, suitability, and density together) outperformed a model focused only on host density and species richness. For each parasite, we compared the models using delta AIC and $R^2$ metrics (conditional and marginal $R^2$).

**Contrasting diversity–disease relationships across scales.** Finally, we compared the influence of host richness on infection success at the two biological scales (i.e., comparing richness effects from the host vs. parasite perspectives). Using the predict function from models at each

scale, we generated model-estimated numbers of metacercariae per host (log-transformed averages among all host species) and total parasite density (the summed products of each host species' average infection load and its density) as a function of host species richness. By overlaying the two predicted curves (infection success in individuals vs. communities), we evaluated how assessments made at the two biological scales altered our perception of diversity–disease relationships. This approach was repeated for each parasite species. We generated predictions at high levels of infection pressure ($\log_{10}$-transformed cercaria density = 2). The host scale model for all amphibian species (see results in Fig. 3) only included infection pressure and host richness as covariates, so we did not generate predictions based on other variables. Similarly, the community scale model only included infection pressure and community competence (and their interaction) as covariates, so we did not generate predictions based on other variables. To generate predictions for each richness value from the community scale model, we first predicted the model at each naturally occurring level of community competence and then appended these predictions to the corresponding richness of the community from which they originated.

### Reporting summary
Further information on research design is available in the Nature Portfolio Reporting Summary linked to this article.

## Data availability
The data generated in this study have been deposited in the Figshare database under https://doi.org/10.6084/m9.figshare.24982794. Source data are provided with this paper.

## Code availability
The code used in this study has been deposited in the Figshare database https://doi.org/10.6084/m9.figshare.24982794.

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

## Acknowledgements

We thank T. McDevitt-Galles, W. Moss, S. Paull, B. Hobart, D. Preston, K. Richgels, and D. Rose for assistance in sampling field sites, and California State Parks, East Bay Regional Parks District, East Bay Municipal Utilities District, Santa Clara County Parks, Open Space Authority, Midpeninsula Open Space, Blue Oaks Ranch Reserve, San Felipe Ranch, and multiple private landowners for facilitating property access. For their contributions to parasite assessments and experiments, we are grateful to J. Bowerman, J. Carder, L. Guderyahn, E. Hannon, B. LaFonte, K. McCaffrey, T. McDevitt-Galles, T. Riepe, and E. Ursich. Dana Calhoun

provided essential contributions to data management, parasite assessment, and project feedback. This research was supported through funding from the National Science Foundation (DEB-0841758, DEB-1149308, IOS-1754886), NSF/NERC Lead Agency Grant (DEB-1754171 and NE/S013369/1), the National Institutes of Health (R01GM109499, R01GM135935), California Department of Fish and Wildlife and US Fish and Wildlife Service (Section 6 P188010), National Geographic Society, and the David and Lucile Packard Foundation. T.E.S.M. was supported as a Simons Foundation fellow of the Life Sciences Research Foundation. Publication of this article was funded by the University of Colorado Boulder Libraries Open Access Fund.

## Author contributions

P.T.J.J. and A.F. designed the study; T.E.S.M. and P.T.J.J. organized the data, developed the analytical framework, and performed the statistical analyses, with feedback from coauthors. P.T.J.J. wrote the first draft, and T.S.M., A.F. and A.D.D. contributed to revising the manuscript.

## Competing interests

The authors declare no competing interests.
