## [Peer Review File · Nature Communications]

Diverging effects of host density and richness across biological scales drive diversity-disease outcomesReviewers' Comments:

Reviewer #1:

Remarks to the Author:

This paper advances the study of how biodiversity affects the transmission, infection intensity, and prevalence of parasites and pathogens and hence key aspects of disease risk. The study focuses on trematode parasites the reproductive stages of which infect aquatic-stage amphibians in ponds and in which snails act as intermediate hosts. The authors have published very prolifically on this system in many journals for over a decade, including small-scale descriptive, experimental, and theoretical components. The current study is largely field-based although lab components are included, consisting of an unusually large number of pond communities over multiple years. By including estimates of abundance and transmission of parasites from intermediate to definitive hosts and by including multiple parasites and hosts, they are able to paint an unusually broad picture of how the host community affects the parasite community.

As in prior papers by this research group, there is much to appreciate in this study, and I think it constitutes one of the most robust assessments of how the host community affects parasite abundance and infection. The writing is generally clear, the data set monumental, and the analytical methods sound. In essence, the research team finds that high host diversity powerfully reduces parasite infection in all the host species and for all the parasite species studied. They have reported these results before, supporting broad patterns also observed in many other ecological systems. In an interesting addition, they found that community assembly was largely additive, meaning that as more host species are included in higher-diversity communities, the total abundance of hosts (of all species) increases. This contrasts with many other communities in which assembly is compensatory (added species reduce abundance of prior species, via competition or predation). Some theory predicts that the negative effect of species richness on pathogen transmission will be detected only in communities with compensatory assembly, and not in those with additive assembly. The authors focus on total community competence remaining stable with increasing host richness (while species-specific competence declined strongly) and point to this as a novel finding helping to reconcile prior debates.

Addressing a few important points would strengthen the paper considerably and increase its likely influence.

The first point is that it is not clear why the authors do not emphasize the important finding that, even with additive community assembly, they observed strong reductions in individual and average host competencies and infection risk for essentially all individuals.

The second point is that the authors should rethink the way they cast the study, as the current Introduction is inaccurate and misleading. The statement starting line 39 (end of first paragraph) mischaracterizes the discussion of the central issue and needs to be replaced. There is no longer debate about whether biodiversity generally decreases pathogen transmission, or about whether biodiversity loss generally increases pathogen transmission. Similarly, there is no debate about the fact that there are exceptions to this generality, sometimes called "idiosyncracies", nor that variability between systems exists. Replacing this language is important in avoiding the perpetuation of a false controversy over whether the dilution effect is general or whether diversity effects vary between systems. The important intellectual issue is understanding the conditions under which dilution effects occur. Such a casting of the study may not be as "sexy" but is more faithful to the science.

The third point is that the term "scale" is not clear and may be a misnomer for what's apparently being represented. A typical interpretation of the term would imply a spatial extent or grain size, and indeed a prior literature asks explicitly about effects of scale on diversity-disease relationships (uncited). I think the authors mean the level of biological organization rather than scale, or perhaps they envision a multidimensional concept of inclusiveness levels, but in any event, the concept requires improved clarity.

The fourth point is that other terms and concepts also require clarity and consistency. I strongly urge the authors to provide glossary, perhaps associated with Fig 1, for this purpose. This should include specifying exactly what was measured, and what was modeled, to provide estimates. It should also include the relevance of the term/concept for actual disease. For instance, "transmission" is estimated as either the average infection load in individual hosts or the total "infection success" across the host community. While it's clear that one difference is the level of organization of the host (individual vs community), it's not clear why the parameter is a load for individuals and "success" for the community, nor how either is relevant to disease. Diverse terms seem to be used for the same concept. For instance, it seems that "total parasite infection success" might mean the total number of parasites, but it's not clear if this differs from "community-wide infection success" or "community-wide transmission success". The Statistical Analysis section refers to "host infection risk" (why not infection intensity, or infection load?) and "parasite infection success" (why not total parasite abundance [and I don't think 'density' should be used, as the area under study is not specified]).

A more minor point, in the opening paragraph of the Discussion, is how the study really differs from the "snapshot" approaches being criticized. Is it not simply a series of snapshots? Is transmission being directly measured as a specified instantaneous rate?

In sum, a revision of the context/introduction of the issues motivating the study, and improved clarity and consistency in the terms used, are crucial. With such changes, the manuscript is likely to be acceptable and indeed strongly influential.

Reviewer #2:

Remarks to the Author:

The manuscript by Johnson et al provides the results of an exhaustive study of parasite transmission in wild amphibian communities, including a combination of extensive field surveys and laboratory assays aimed at understanding how host communities influence the spread of parasites. Overall, the study shows no clear relationship between diversity and disease. Instead, the study shows strong evidence that parasite transmission (measured as the relationship between infection pressure, a composite of several snail characteristics measured early in the season, and disease measured in amphibians later in the season), is strongly influenced by host richness, but only at the scale of individual host species, and not at the scale of whole host communities, which the authors attribute to shifts in the density of the most competent hosts with increasing host richness. Using a single measure of host community competence computed from experimental assays in the lab provides a more parsimonious explanation of overall infection (compared to a global model including richness, density, and parasite predators), though quantitative comparison of predictive performance or effects on parasite transmission are difficult to interpret. The authors conclude that these results will help reconcile diversity-disease debates by focusing on various ecological drivers of parasite transmission across biological scales rather than providing limited "snapshots" of infection in space and time.

Overall, I am impressed with this study, but I feel like the authors need to improve the clarity and presentation of data and results and to more clearly define and test parasite transmission.

Major comments:

Presentation of data and results. I recognize that this manuscript includes a large number of statistical tests and approaches, and many many results. However, I felt that the Results section was overly interpretive, and it seemed inappropriate that few values from statistical tests were actually reported in the main text. I would suggest that if a result is sufficiently important to interpret in the Results section, then that text should be supported with some reference to model estimates/effect sizes, statistical tests, and goodness of fit tests (which should also be reported in the text, if possible). An

added bonus to this is that having specific values in the text makes it MUCH easier to find relevant results in the supplemental tables.

Similarly, the analyses were poorly articulated in the manuscript, and often involved the use of values that were computed from several different sources, with poor explanation of how these values were computed and how sensitive the analyses were to this approach. I would encourage the authors, in future submissions, to provide reproducible code and data to assist reviewers in assessing the robustness of the analytical approach.

Definition of transmission. A core focus of this manuscript is parasite transmission, but there is no clear operational definition of "transmission" until the statistical analysis subsection of the Methods. An operational definition of "transmission" should be provided much earlier in the text to aid with interpretation of the Results. After reading the manuscript in full, it became clear that transmission was estimated as the relationship between a composite variable, termed "snail infection pressure", measured early in the season, and amphibian infection intensity measured later in the season. Looking at how the slope of this relationship changes with increasing richness seems like a very strong and compelling approach to assess diversity-disease relationships. Nevertheless, this operational definition of transmission (assuming that I got it correct) needs to be unpacked in more detail in the Introduction and Results sections.

I also have some concerns about the measurements used to compute infection pressure. Clearly, a lot of work went into computing this variable, including data from both field and laboratory assays. Yet, I was disappointed that so little of the main text involved describing this key variable. Furthermore, I have some concern about the sensitivity of the analyses to variation in infection pressure. As an example, Table S5 shows considerable variation in the relationship between snail size and cercarial release. How sensitive are the models to variation around this mean? Similarly, only a small number of snails were surveyed in each pond, how sensitive are the analyses to variation in measured snail density and snail prevalence

Comparison of competence to 'global model'. I was confused about the model selection presented in Table S4, and the degree to which it supported the manuscript. As far as I can tell, this analysis was a comparison of a model with only two main effects and one interaction term to a model with four main effects and three interaction terms, many of which were previously shown to have limited statistical support in previous analyses. So it came as no surprise that the models with community competence had lower AICs than the global models. But how does this show that community competence is a better predictor/driver of infection/transmission than richness or density? And why did this analysis compare a competence model to a full global model, when other analyses presented reduced versions of that same global model? As far as I can tell, the AIC model selection suggests that community competence and its interaction with infection pressure, provide a more parsimonious explanation of disease than richness, density, and predators (and their interactions with infection pressure). However, this approach does not clearly show that community competence is a better predictor of disease than diversity. I was also confused about whether this model comparison provides any evidence in support of the manuscripts' focus on transmission, or if it is instead asking a separate question about overall infection intensity? I think the authors need to more clearly articulate what this model comparison can and cannot test, and carefully revise their language regarding its interpretation. The approach could be further improved by providing the model fit statistics for the other models (and the global model?) in the paper so that readers could see whether the community competence model explains more overall variation in infection intensity than the other models.

Minor comments

The introduction is beautifully written and quite easy to follow. I especially appreciated the "host" vs "parasite" perspective that the authors linked to scale. I applaud the authors for taking such a complicated and difficult subject and making it so easily digestible. Because the manuscript is integrating several areas of past research into a single framework, there is a tremendous literature on

individual topics covered in the introduction that unfortunately goes uncited, presumably due to page-length or citation limits. This is a missed opportunity, but I suppose that an interested reader could look into the recent reviews and syntheses cited in this study to identify relevant primary studies on their own if needed.

The authors used a model reduction approach to model selection, but also noted that all variables were standardized and centered and did not suffer from collinearity. Could the authors briefly justify this model selection approach and describe how sensitive the results are to this method? Similarly, it would be helpful if the authors could also provide the output from the full models, since data and code are not provided to readers or reviewers.

As noted in the introduction, many of the variables considered in these analyses can co-vary with one another under natural conditions. The authors briefly mention that variables were not collinear ($r < 0.7$), but do not provide any information about correlations among predictor variables (and their interactions). How strongly linked are host density, richness, competence, and infection pressure and how sensitive are the models to covariance among these factors (and their interactions)? Ideally, this information would be included in the supplement, since the data and code are not available to readers or reviewers.

Several previous studies (including many from this system) suggest that different drivers of community assembly or disassembly can influence diversity-disease relationships in different ways. I would be curious to know what processes are thought to drive assembly or disassembly of amphibian communities in this system (why are some ponds more species rich than others?), and how that might influence the patterns of covariance between host richness, density, and competence that were observed?

Line 20 – I do not see any statistical support for this statement, as the manuscript does not present any main effect of richness on infection risk that are supported by the model ($p < 0.05$). Do the authors mean “transmission” here?

Line 80 –Already in the manuscript, it would be helpful for readers to know that the variables in this model are mean-centered.

Line 81: This term, “infection pressure”, should be clearly defined the first time it shows up in the manuscript. As currently written, the only explanation of this variable in the main text shows up at Line 311 (how this value is actually computed is in the SI, presumably due to page limitations). This variable is the core of the manuscript, and readers need to be able to understand what it is, how it was computed, and how sensitive analyses are to its variation.

Line 98: unclear what is meant by “all host individuals”. Please clarify that these data are weighted by the relative abundance of each host species in the community (I believe this is mentioned in the statistical analysis subsection of the Methods).

Line 105: I think the authors meant to refer to Table S3 here, or else am I missing something important in the text?

Line 114: Unclear why this value is not compared to a similar value from the global model / other models in the manuscript.

Line 127 – if density and richness are non-independent, does this suggest that they should not be included in the same model (e.g., all previous models presented in the MS)? The authors note that these values are not collinear, but how sensitive are the models to inclusion of each variable in each model?

Line 128 – I am not sure that I understand this statement, but it seems critical to the manuscript. Could this be explained more clearly, and perhaps supported with numerical results?

Line 153: Could the authors unpack why certain measures of disease are expected to be inappropriate for understanding diversity-disease relationships? I understand that this manuscript focuses on relationships between diversity and transmission, but it's less clear to me why the results of transmission are better than so-called "snapshot" measures of infection? In this system, transmission occurs from snails to amphibians, and not among amphibian hosts, but this seems quite different from other commonly studied systems, which might involve both inter and intraspecific transmission among hosts.

Line 179 – I worry that the authors are conflating transmission in this paragraph (which was modeled as the relationship between pressure and infection risk) with infection risk, which is a term the authors use to describe observed infection (Figure 1). As far as I can tell from the models, there is no statistical evidence that increasing richness broadly reduced individual host infection risk. Increasing richness broadly reduced transmission (changed the slope of the pressure-risk relationship), but I do not see any results that support the idea of a net negative effect on infection risk. I think that this is really important – because while this study shows strong effects of diversity on transmission, I do not see any results that clearly document a net relationship between diversity and disease, per se.

Line 310: Here and elsewhere in the paper, it would be helpful to spell out exactly what value is being interpreted as "parasite transmission". My understanding is that transmission is the relationship between log₁₀-transformed "infection pressure" and the various responses in the model.

Line 334 – I do not see these results presented anywhere.

Line 342 – by what were the predictors scaled?

Line 344 – What is the justification for this model-selection approach?

Line 363 – Please describe this measurement earlier in the text, where the variable is first defined. It was unclear until now that this value controlled for host density.

Figure 2. How were the values for the black and grey lines (1 spp and 4+ spp) selected? And how were these lines fit? Do these estimates come from the model presented in Table S1? Or do these lines represent some other analysis of the data? I guess the most appropriate way to fit this would be with a post-hoc analysis of the models provided in Table S1 (i.e., a simple-slopes analysis or something similar). Also a bit confused where the coefficient estimates for the inlays come from, since these values are not reported in Table S1 or in the text. Are these from the full model?

Figure 3. Please clarify where these model coefficients come from. Note that there are no coefficients provided in Table S2, and so I cannot look them up to see what they refer to. My best guess is that this is the Pressure X Richness coefficient estimated separately from each host species (so coming from the three-way interactions)? I find it confusing that the language in this figure does not match up at all with the language in the Results or in Figure 2. Dilution and amplification are not mentioned at all in the results, and it is very difficult to figure out the link between the interaction between infection pressure and species richness and the richness effect on transmission, since there is no operational definition of "transmission". It is clear from reading the statistical analysis subsection of the Methods section that the effect of infection pressure on infection load is being interpreted as transmission, and so the interaction between this effect and richness is the "dilution" or "amplification" effect, but the link between this figure and the actual text in the Results and Methods needs to be strengthened.

Figure 4. Please clarify that these predictions are being generated at mean values for the other variables in these models, rather than at 0.

Table S1, please specify in the legend how these variables were transformed (i.e., infection pressure is log-10 transformed), including standardization and mean-centering. Where do these p-values come from? Are they from a likelihood ratio test? Are the full (i.e., not reduced) model results presented anywhere? Do these models also have an estimated intercept? It is also frustrating that parasite names are not consistent with the main text (e.g., *Alaria* in Figure 2 vs *A. marcinae* in Table S1).

Responses to reviewers

Referees' comments to the author(s):

Reviewer #1 (Remarks to the Author):

This paper advances the study of how biodiversity affects the transmission, infection intensity, and prevalence of parasites and pathogens and hence key aspects of disease risk. The study focuses on trematode parasites the reproductive stages of which infect aquatic-stage amphibians in ponds and in which snails act as intermediate hosts. The authors have published very prolifically on this system in many journals for over a decade, including small-scale descriptive, experimental, and theoretical components. The current study is largely field-based although lab components are included, consisting of an unusually large number of pond communities over multiple years. By including estimates of abundance and transmission of parasites from intermediate to definitive hosts and by including multiple parasites and hosts, they are able to paint an unusually broad picture of how the host community affects the parasite community.

As in prior papers by this research group, there is much to appreciate in this study, and I think it constitutes one of the most robust assessments of how the host community affects parasite abundance and infection. The writing is generally clear, the data set monumental, and the analytical methods sound. In essence, the research team finds that high host diversity powerfully reduces parasite infection in all the host species and for all the parasite species studied. They have reported these results before, supporting broad patterns also observed in many other ecological systems. In an interesting addition, they found that community assembly was largely additive, meaning that as more host species are included in higher-diversity communities, the total abundance of hosts (of all species) increases. This contrasts with many other communities in which assembly is compensatory (added species reduce abundance of prior species, via competition or predation). Some theory predicts that the negative effect of species richness on pathogen transmission will be detected only in communities with compensatory assembly, and not in those with additive assembly. The authors focus on total community competence remaining stable with increasing host richness (while species-specific competence declined strongly) and point to this as a novel finding helping to reconcile prior debates.

AUTHOR RESPONSE: We appreciate the thoughtful feedback and insightful suggestions for improvement. In response, we have clarified the terminology throughout the paper (including the addition of a glossary, as suggested), revised the Introduction to emphasize the need for a predictive framework, and taken care to indicate our focus on biological level of organization (rather than spatial scale per se).

Addressing a few important points would strengthen the paper considerably and increase its likely influence.

The first point is that it is not clear why the authors do not emphasize the important finding that, even with additive community assembly, they observed strong reductions in individual and average host competencies and infection risk for essentially all individuals.

AUTHOR RESPONSE: Good suggestion. We have made this point clearer in the revised manuscript, including more detailed information on the links among community assembly, competence, and transmission both in the Introduction and Discussion.

“In particular, dilution effects may be expected to occur more often when community assembly is substitutive as opposed to additive^{2,11}. In the former case, individuals of added species replace those already present and total host density remains constant. If newly added individuals are of lower average competence than the individuals they replace, overall transmission is likely to decrease via dilution effects. When assembly is additive, however, increases in richness covary positively with increased host densities, because individuals of new species are added to those of the existing community; hence overall transmission may increase with host richness¹⁵.” (see lines 53-60)

“It is noteworthy that this diluting effect was evident even though communities assembled additively (i.e., density increased with richness), rather than substitutively (in which added individuals replace existing ones)^{2,24}.” (see lines 213-215)

“Theory (and previous empirical data) suggest that the negative effects of species richness on pathogen transmission are most likely to manifest in communities with substitutive (compensatory) assembly, in which individuals of newly added species are associated with decreases in the density of existing species (e.g., through competition or predation). If added individuals have lower average parasite competence than the individuals they replace, we would expect to see reductions in transmission. Conversely, communities that assemble additively with an increase in total host density with increasing species richness are expected to show either weak or positive relationships between richness and transmission or disease (especially over large spatial or temporal scales).” (see lines 234-242)

The second point is that the authors should rethink the way they cast the study, as the current Introduction is inaccurate and misleading. The statement starting line 39 (end of first paragraph) mischaracterizes the discussion of the central issue and needs to be replaced. There is no longer debate about whether biodiversity generally decreases pathogen transmission, or about whether biodiversity loss generally increases pathogen transmission. Similarly, there is no debate about the fact that there are exceptions to this generality, sometimes called “idiosyncracies”, nor that variability between systems exists. Replacing this language is important in avoiding the perpetuation of a false controversy over whether the dilution effect is general or whether diversity effects vary between systems. The important intellectual issue is understanding the conditions under which dilution effects occur. Such a casting of the study may not be as “sexy” but is more faithful to the science.

AUTHOR RESPONSE: We have completely recast the Introduction to address this suggestion. It now emphasizes that substantial evidence supports the link between biodiversity and disease risk, and that contemporary challenges center around the development of a more predictive framework for explaining heterogeneity in such effects among systems.

For instance, from the Introduction:

“While initial debates on diversity-disease relationships centered around whether biodiversity generally protects against disease (via the ‘dilution effect’), more recent syntheses have established firmly that biodiversity often inhibits parasite transmission but that there exists considerable variability in the occurrence and magnitude of this effect among systems⁶⁻¹¹. The challenge now is to understand the drivers of such heterogeneity across systems and delineate under what host, parasite, and environmental conditions changes in diversity amplify or dilute disease risk^{12,13}.” (see lines 38-44).

We have also rephrased the title to move away from language relating to ‘debates’.

The third point is that the term “scale” is not clear and may be a misnomer for what’s apparently being represented. A typical interpretation of the term would imply a spatial extent or grain size, and indeed a prior literature asks explicitly about effects of scale on diversity-disease relationships (uncited). I think

the authors mean the level of biological organization rather than scale, or perhaps they envision a multidimensional concept of inclusiveness levels, but in any event, the concept requires improved clarity.

AUTHOR RESPONSE: This is an excellent point. We now make it clear that our focus is on “biological scale” and the level of organization (i.e., individual hosts vs. the host community), rather than spatial scale (grain and extent). Incorporation of a glossary in response to the next point has also helped to formalize the terminology.

“Second, the effects of diversity on parasite transmission can have opposing effects depending on biological scale¹⁷: how community composition influences infection of individual hosts may differ from how it affects total infection success for the parasite across the available host community (Fig. 1).” (see lines 61-64).

The fourth point is that other terms and concepts also require clarity and consistency. I strongly urge the authors to provide glossary, perhaps associated with Fig 1, for this purpose. This should include specifying exactly what was measured, and what was modeled, to provide estimates. It should also include the relevance of the term/concept for actual disease. For instance, “transmission” is estimated as either the average infection load in individual hosts or the total “infection success” across the host community. While it’s clear that one difference is the level of organization of the host (individual vs community), it’s not clear why the parameter is a load for individuals and “success” for the community, nor how either is relevant to disease. Diverse terms seem to be used for the same concept. For instance, it seems that “total parasite infection success” might mean the total number of parasites, but it’s not clear is this differs from “community-wide infection success” or “community-wide transmission success”. The Statistical Analysis section refers to “host infection risk” (why not infection intensity, or infection load?) and “parasite infection success” (why not total parasite abundance [and I don’t think ‘density’ should be used, as the area under study is not specified]).

AUTHOR RESPONSE: The addition of a glossary to the revised manuscript has greatly helped to clarify the terminology and analyses. We have also used this opportunity to comb through the paper and standardize word use. As part of this revision, we make it clear that all of our analyses focused on transmission, i.e., the capacity of infective parasites emerging from snails to find and infect amphibian hosts. So our response variable is infection load within amphibian hosts, considered either at the individual host level (infection load per host) or across the entire amphibian host community (total parasite abundance). We note that this latter response is still a density given that it is total parasite abundance per netsweep (i.e., our unit of abundance for the amphibian hosts is per dipnet sweep).

A more minor point, in the opening paragraph of the Discussion, is how the study really differs from the “snapshot” approaches being criticized. Is it not simply a series of snapshots? Is transmission being directly measured as a specified instantaneous rate?

AUTHOR RESPONSE: The point of emphasis here is that our study focuses on transmission between hosts (in this case, between infected snails and suitable amphibian hosts). While transmission (between hosts or through time) is often more difficult to measure in field systems, we contend that it offers a more direct parallel both to theoretical work and to experimental studies. Had we simply measured infection load in amphibians, without incorporating information about infection pressure from snail intermediate hosts, we would have failed to detect the considerable effect of host richness on the system. Thus, our emphasis is on process (transmission) rather than on pattern (measurement of infection without consideration of infection pressure). Given that many diseases involve multiple hosts/vectors and likely exhibit non-equilibrium dynamics, we suggest that

greater effort to measure transmission empirically is essential for linking patterns in the field to theory and experiments.

We have updated both the Introduction and Discussion to make this distinction more transparent, for instance:

“We focused specifically on how diversity affects parasite transmission (Glossary), operationally defined here as the slope of the relationship between the number of trematode infective stages (cercariae) and the number of established parasites in amphibian hosts, rather than on static measures of infection levels in the hosts. Dilution effects generally act through changes to the process of transmission, for example through encounter reduction⁵; thus, although often more difficult to measure in field systems, evaluating changes in transmission offers a more direct parallel to predictions from theory and experimental studies.” (lines 81-88).

In sum, a revision of the context/introduction of the issues motivating the study, and improved clarity and consistency in the terms used, are crucial. With such changes, the manuscript is likely to be acceptable and indeed strongly influential.

AUTHOR RESPONSE: Again, we deeply thank the reviewer for the constructive suggestions, which we believe have greatly improved the manuscript.

Reviewer #2 (Remarks to the Author):

The manuscript by Johnson et al provides the results of an exhaustive study of parasite transmission in wild amphibian communities, including a combination of extensive field surveys and laboratory assays aimed at understanding how host communities influence the spread of parasites. Overall, the study shows no clear relationship between diversity and disease. Instead, the study shows strong evidence that parasite transmission (measured as the relationship between infection pressure, a composite of several snail characteristics measured early in the season, and disease measured in amphibians later in the season), is strongly influenced by host richness, but only at the scale of individual host species, and not at the scale of whole host communities, which the authors attribute to shifts in the density of the most competent hosts with increasing host richness. Using a single measure of host community competence computed from experimental assays in the lab provides a more parsimonious explanation of overall infection (compared to a global model including richness, density, and parasite predators), though quantitative comparison of predictive performance or effects on parasite transmission are difficult to interpret. The authors conclude that these results will help reconcile diversity-disease debates by focusing on various ecological drivers of parasite transmission across biological scales rather than providing limited “snapshots” of infection in space and time.

Overall, I am impressed with this study, but I feel like the authors need to improve the clarity and presentation of data and results and to more clearly define and test parasite transmission.

AUTHOR RESPONSE: Thank you to the Reviewer for the feedback and suggestions for improvement. These suggestions have helped us to strengthen the manuscript. More specifically, we have incorporated more of the statistical results into the manuscript (rather than only in the SI), we have made the terminology clearer and more consistent (including addition of a glossary), and we have added additional detail about how we measure transmission to the main text (specifically into the Introduction and Results).

Major comments:

Presentation of data and results. I recognize that this manuscript includes a large number of statistical tests and approaches, and many many results. However, I felt that the Results section was overly interpretive, and it seemed inappropriate that few values from statistical tests were actually reported in the main text. I would suggest that if a result is sufficiently important to interpret in the Results section, then that text should be supported with some reference to model estimates/effect sizes, statistical tests, and goodness of fit tests (which should also be reported in the text, if possible). An added bonus to this is that having specific values in the text makes it MUCH easier to find relevant results in the supplemental tables.

AUTHOR RESPONSE: As part of the revised manuscript, we have moved many of the statistical result values (coefficients, standard errors, and P -values) to the main text. We still maintain the full result tables in the supplement, facilitating cross-referencing between the two documents as suggested. We also include additional diagnostic assessments for GLMMs in the paper, such as the conditional and marginal R^2 values as well as tests for overdispersion and collinearity (i.e., variance inflation factors). Data and code for reproducing the analyses are further included with this submission.

Similarly, the analyses were poorly articulated in the manuscript, and often involved the use of values that were computed from several different sources, with poor explanation of how these values were computed and how sensitive the analyses were to this approach. I would encourage the authors, in future submissions, to provide reproducible code and data to assist reviewers in assessing the robustness of the analytical approach.

AUTHOR RESPONSE: The revisions include enhanced details on the analyses alongside reproducible code for use with the data. We hope this helps to address this shortcoming of the original submission. We also include further details about the measurements of transmission, as further articulated in responses below.

Definition of transmission. A core focus of this manuscript is parasite transmission, but there is no clear operational definition of “transmission” until the statistical analysis subsection of the Methods. An operational definition of “transmission” should be provided much earlier in the text to aid with interpretation of the Results. After reading the manuscript in full, it became clear that transmission was estimated as the relationship between a composite variable, termed “snail infection pressure”, measured early in the season, and amphibian infection intensity measured later in the season. Looking at how the slope of this relationship changes with increasing richness seems like a very strong and compelling approach to assess diversity-disease relationships. Nevertheless, this operational definition of transmission (assuming that I got it correct) needs to be unpacked in more detail in the Introduction and Results sections.

AUTHOR RESPONSE: Great point, and one we have embraced whole-heartedly as part of these revisions. We have worked to refine and standardize the terminology in the paper and now include a glossary that addresses this point directly. We also have incorporated an operational definition of transmission into the manuscript in several locations, including most prominently in the Introduction, as suggested.

“We focused specifically on diversity effects on parasite transmission, operationally defined here as the slope of the relationship between the number of trematode infective stages (cercariae) and the number of established parasites in amphibian hosts, rather than on static snapshots of infection levels in the hosts. Dilution effects generally act through changes to the process of transmission, for example through encounter reduction (Keesing et al. 2006); thus, although often more difficult to measure in

field systems, evaluating changes in transmission offers a more direct parallel to predictions from theory and experimental studies.” (see lines 81-88).

I also have some concerns about the measurements used to compute infection pressure. Clearly, a lot of work went into computing this variable, including data from both field and laboratory assays. Yet, I was disappointed that so little of the main text involved describing this key variable. Furthermore, I have some concern about the sensitivity of the analyses to variation in infection pressure. As an example, Table S5 shows considerable variation in the relationship between snail size and cercarial release. How sensitive are the models to variation around this mean? Similarly, only a small number of snails were surveyed in each pond, how sensitive are the analyses to variation in measured snail density and snail prevalence

AUTHOR RESPONSE: Agreed. We have now added additional text both to the Introduction and to the Methods of the main text dealing with this variable in greater detail. We consider it a major strength of the analysis to include an estimate for the density of infective stages in the system, which is something often simplified to infection prevalence or infected host density. By further incorporating the influence of snail host size, which is frequently associated with cercariae production and release, we can explain significantly more variation than when using infected snail density alone.

We exercised care early on in our choice of metric for “infection pressure”, and initially examined three potential options: prevalence of infection in snails, density of infected snails, and infected cercaria densities. While the reviewer is correct that this third option can propagate error (as a compound term), it represented our best attempt at accurately estimating exposure (a notoriously challenging parameter to measure). We closely examined which of these metrics was the best predictor of infection prior to running any models that tested hypotheses with diversity-related measures. In our examination, we regressed each metric against infection load in chorus frogs (our indicator species) for all four parasite species and then compared the fit of the resulting models with AIC. Infected cercaria densities (our compound metric) dramatically outperformed the other metrics, and visual inspections showed a strong fit of this predictor to observed infections. We now include the results of this exploration (both as a table and figure) in the SI. As for snail sample size, we generally sampled 100 snails per pond per year and per species, which is a commonly used sample size in the study of trematode infections. In total, more than 200,000 snails were dissected.

SI additional explanatory text:

“Formulation of predictor variables

***Infection pressure:** To estimate amphibian exposure to infective cercariae, we developed three alternative metrics representing ‘infection pressure’: prevalence of infection in snails, density of infected snails, and estimated density of infective cercariae. Prevalence of infection represents the number of infected snails divided by the number dissected. Density of infected snails was estimated as the product of snail density (average number captured per dipnet sweep) and the prevalence of snails exhibiting infection. Our estimate of cercaria densities included two additional pieces of information. Specifically, we multiplied the average size of dissected snails in a pond by a regression equation relating snail size to cercariae release over 24 hours (as quantified from a subset of infected snails for each trematode under standardized conditions, see Table S6). This relationship recognizes that the number of cercariae emerging from infected snails varies substantially as a function of snail size¹⁴. The product of terms – snail density, infection prevalence, mean size and the size-specific slope of cercariae release – was used to generate estimates of cercaria density for each trematode species and site-by-year observation. We determined which of these estimates served as the best proxy for exposure by regressing them against observed infections in *P. regilla* (our indicator species) and comparing model fit with AIC. These models took the same form as transmission models at the individual host*

*scale, but without any factors representing hypotheses related to diversity. With the function `glmmTMB`¹⁵, we ran generalized linear mixed effects models (overdispersed Poisson distribution) with total metacercariae observed in a pond as the response variable and an offset term for the number of amphibians dissected. Our estimate of infection pressure was the only fixed effect, and we included random effects for site, year, and an observation-level random effect for the site-year combination. These preliminary analyses indicated that the estimated density of infective cercariae was by far the best predictor of infection success in *P. regilla* (Table S7 and Fig. S1), and we therefore used this proxy to represent “infection pressure” in all subsequent models.” (see SI lines 62-86).*

Comparison of competence to ‘global model’. I was confused about the model selection presented in Table S4, and the degree to which it supported the manuscript. As far as I can tell, this analysis was a comparison of a model with only two main effects and one interaction term to a model with four main effects and three interaction terms, many of which were previously shown to have limited statistical support in previous analyses. So it came as no surprise that the models with community competence had lower AICs than the global models. But how does this show that community competence is a better predictor/driver of infection/transmission than richness or density? And why did this analysis compare a competence model to a full global model, when other analyses presented reduced versions of that same global model? As far as I can tell, the AIC model selection suggests that community competence and its interaction with infection pressure, provide a more parsimonious explanation of disease than richness, density, and predators (and their interactions with infection pressure). However, this approach does not clearly show that community competence is a better predictor of disease than diversity. I was also confused about whether this model comparison provides any evidence in support of the manuscripts’ focus on transmission, or if it is instead asking a separate question about overall infection intensity? I think the authors need to more clearly articulate what this model comparison can and cannot test, and carefully revise their language regarding its interpretation. The approach could be further improved by providing the model fit statistics for the other models (and the global model?) in the paper so that readers could see whether the community competence model explains more overall variation in infection intensity than the other models.

AUTHOR RESPONSE: Apologies for the confusion here. Based on these suggestions, we modified these model comparisons in two ways. First, we omit aquatic predators from either model, such that we are now doing a more direct comparison of a model that includes host richness and density as separate variables versus one that focuses on ‘competence’ (an integrated measure of host density, species composition, and suitability). This acknowledges that the role of predators is really a distinctly different mechanism from those directly involving hosts. Second, we incorporate additional measures of model fit such as conditional and marginal R^2 for each model, thereby facilitating comparison. We explain this approach in further detail within the Methods:

“When considering parasite infection success across the host community, interspecific differences in host suitability also matter; transmission is a function not only of whether a parasite can contact a host, but also whether it can successfully infect it. We therefore incorporated community-level estimates of community competence, d_c , which simultaneously integrate information about host density, species composition, and the functional suitability of hosts for supporting infection (see ‘Experimental infections to quantify competence’). For each parasite species, we compared the explanatory power of a model containing host density, host richness, and their interactions with infection pressure, with a model including community competence and its interaction with infection pressure. Through this comparison, we aimed to assess the degree to which a model comprising an aggregate measure of overall community competence (integrating host identity, suitability and density together) outperformed a model focused only on host density and species richness. For each parasite, we compared the models using delta AIC and R^2 metrics (conditional and marginal R^2).” (see lines 444-456).

In addition, we also calculated the AIC values for simpler models that only included infection pressure and host density (main effects and interaction) or infection pressure and host richness (main effects and interaction) for each of the four parasites. These models consistently exhibited higher AIC scores than the model involving host competence and its interaction with infection pressure: *Alaria* ($\Delta\text{AIC} = -27.3$ and -45.9 , respectively); *Cephalogonimus* ($\Delta\text{AIC} = -21.6$ and -134.7), *Echinostoma* ($\Delta\text{AIC} = -17.7$ and -113.1), and *Ribeiroia* ($\Delta\text{AIC} = -5$ and -24 , respectively). Currently these models are not included in the SI, but we are happy to incorporate them if they help to further address this point.

Minor comments

The introduction is beautifully written and quite easy to follow. I especially appreciated the “host” vs” “parasite” perspective that the authors linked to scale. I applaud the authors for taking such a complicated and difficult subject and making it so easily digestible. Because the manuscript is integrating several areas of past research into a single framework, there is a tremendous literature on individual topics covered in the introduction that unfortunately goes uncited, presumably due to page-length or citation limits. This is a missed opportunity, but I suppose that an interested reader could look into the recent reviews and syntheses cited in this study to identify relevant primary studies on their own if needed.

AUTHOR RESPONSE: We have broadly re-written the Introduction in response to comments from Reviewer 1, and now also have added additional citations to help provide readers with further background on diversity-disease research.

The authors used a model reduction approach to model selection, but also noted that all variables were standardized and centered and did not suffer from collinearity. Could the authors briefly justify this model selection approach and describe how sensitive the results are to this method? Similarly, it would be helpful if the authors could also provide the output from the full models, since data and code are not provided to readers or reviewers.

AUTHOR RESPONSE: After some consideration, we elected to remove the model reduction approach for model selection and instead present the full, global models for each parasite. Results are broadly unchanged, but this allows readers to more directly assess the influence (or lack thereof) for each variable. Outputs from these full models are now included, both in the main text and in the supplementary materials. As an additional note, data and code are provided with this submission.

As noted in the introduction, many of the variables considered in these analyses can co-vary with one another under natural conditions. The authors briefly mention that variables were not collinear ($r < 0.7$), but do not provide any information about correlations among predictor variables (and their interactions). How strongly linked are host density, richness, competence, and infection pressure and how sensitive are the models to covariance among these factors (and their interactions)? Ideally, this information would be included in the supplement, since the data and code are not available to readers or reviewers.

AUTHOR RESPONSE: We now provide correlation tables for our hypothesized predictors as part of the supplementary tables (Table S8). Beyond pairwise correlations, we also calculated the variance inflation factors for each model as a robust way of ensuring a lack of collinearity. These values were generally below 1.5, indicating a low level of collinearity. As an additional note, data and code are provided with this submission.

Several previous studies (including many from this system) suggest that different drivers of community assembly or disassembly can influence diversity-disease relationships in different ways. I would be

curious to know what processes are thought to drive assembly or disassembly of amphibian communities in this system (why are some ponds more species rich than others?), and how that might influence the patterns of covariance between host richness, density, and competence that were observed?

AUTHOR RESPONSE: The current results help to illustrate that average host competence values decrease with host richness, indicating that amphibian community assembly generally involves the progressive addition of less-competent host species. However, assembly also appears to be additive or slightly saturating, such that total host density also climbs with richness. This helps to account for the seemingly discordant effects of richness on parasite transmission when considered at the individual host level versus the entire host community. We are actively working to better understand the processes driving assembly and disassembly in this system, including especially what controls richness across space and through time. This is the subject of an in-progress manuscript.

Line 20 – I do not see any statistical support for this statement, as the manuscript does not present any main effect of richness on infection risk that are supported by the model ($p < 0.05$). Do the authors mean “transmission” here?

AUTHOR RESPONSE: Yes. We have revised this section to ensure the terminology is clearer. We have also incorporated a glossary that helps to avoid confusion on this point.

Line 80 –Already in the manuscript, it would be helpful for readers to know that the variables in this model are mean-centered.

AUTHOR RESPONSE: Good point. We now add this information at the beginning of the Results, as suggested. See line 109.

Line 81: This term, “infection pressure”, should be clearly defined the first time it shows up in the manuscript. As currently written, the only explanation of this variable in the main text shows up at Line 311 (how this value is actually computed is in the SI, presumably due to page limitations). This variable is the core of the manuscript, and readers need to be able to understand what it is, how it was computed, and how sensitive analyses are to its variation.

AUTHOR RESPONSE: In the revised manuscript, we address this definition more directly both at the end of the Introduction and in the Glossary. We have also moved some of the text from supplement into the main text of the paper to address this point.

“We focused specifically on how diversity affects parasite transmission, operationally defined here as the slope of the relationship between the number of trematode infective stages (cercariae) and the number of established parasites in amphibian hosts, rather than on static measures of infection levels in the hosts. Dilution effects generally act through changes to the process of transmission, for example through encounter reduction; thus, although often more difficult to measure in field systems, evaluating changes in transmission offers a more direct parallel to predictions from theory and experimental studies.” (see lines 81-88).

“...infection pressure from snail intermediate hosts (estimated density of infective trematode cercariae in a given pond, quantified from the density of infected snails, average snail size and size-adjusted number of cercariae released based on snail length-to-cercariae regressions; see SI).” (see lines 105-108).

“Thus, all models included a term for ‘infection pressure’ related to the density of trematode cercariae released from snail intermediate hosts per day. Because this value is difficult to measure directly, we combined information on snail infection prevalence, snail density, snail average size, and the relationship between cercarial release and snail size to create a proxy term. We calculated the density of infected snails as the product of snail density (average number captured per dipnet sweep) and the prevalence of snails exhibiting infection (number infected divided by number dissected). Recognizing that the number of cercariae emerging from an infected snail can vary substantially as a function of snail size, we used species-specific regression equations to link average snail size at a site to the expected number of cercariae released over 24 hours (as quantified from a subset of infected snails for each trematode species under standardized conditions, see Table S6). The product of these terms – snail density, infection prevalence, mean size and the size-specific slope of cercariae release – were used to generate estimates of cercaria density for each trematode species and site-by-year observation (see SI for additional details). Preliminary investigations indicated that this proxy better fit observed infection data than either infection prevalence in snails or the density of infected snails alone (Table S7). After log₁₀-transformation, the resulting estimate of infection pressure was included in all models as a predictor of amphibian infection at both scales.” (see lines 372-388).

Line 98: unclear what is meant by “all host individuals”. Please clarify that these data are weighted by the relative abundance of each host species in the community (I believe this is mentioned in the statistical analysis subsection of the Methods).

AUTHOR RESPONSE: We apologize for the lack of clarity, and emphasize that these data are not weighted by relative abundance. Total parasite density is the sum of products of each species’ average infection multiplied by its density. Here, density is an absolute (not relative) value – such that the metric reflects the total number of metacercariae that occur in a dipnet. We have reworded this as:

“Total parasite density (quantified as the sum of each host species’ average infection load multiplied by its larval density) was positively influenced by total host density and negatively affected by the interaction between infection pressure and host richness.” (see lines 139-142).

Line 105: I think the authors meant to refer to Table S3 here, or else am I missing something important in the text?

AUTHOR RESPONSE: Thank you for catching this typo. It has been corrected.

Line 114: Unclear why this value is not compared to a similar value from the global model / other models in the manuscript.

AUTHOR RESPONSE: We have now added the marginal and conditional R^2 values for each model to the paper, including a direct comparison of the global models to the competence models referenced in this section.

Line 127 – if density and richness are non-independent, does this suggest that they should not be included in the same model (e.g., all previous models presented in the MS)? The authors note that these values are not collinear, but how sensitive are the models to inclusion of each variable in each model?

AUTHOR RESPONSE: As part of the revisions, we have further tested for evidence of collinearity using estimation of the variance inflation factors. Based on consistently low VIFs between these two variables (<1.5), it is statistically appropriate to include these terms together in the same model. Thus, while these terms are certainly related, there is considerable variation in the relationship

afforded by the broad range of communities sampled, such that the coefficients for each predictor are not strongly altered by the presence of the other within the model. Also note that at the level of individual hosts, we are examining how the density of an individual host species (in this case, chorus frogs) varies with host richness. The lack of a strong relationship here is consistent with our evidence that communities assemble additively.

Line 128 – I am not sure that I understand this statement, but it seems critical to the manuscript. Could this be explained more clearly, and perhaps supported with numerical results?

AUTHOR RESPONSE: The original sentence was written as: *“Hence, the increases in transmission driven by increases in total host density counteracted any directly negative effects of host richness, resulting in negligible changes on community-wide infection success.”*

We have rewritten the sentence for clarity and now point to the supporting statistical results that are now fully specified in the results.

“Hence, at the community scale, the increases in transmission that resulted from increases in total host density were balanced against any inhibitory effects of host richness (see statistical results in ‘Infection success at the community scale’ and Table S3).” (see lines 171-174).

Line 153: Could the authors unpack why certain measures of disease are expected to be inappropriate for understanding diversity-disease relationships? I understand that this manuscript focuses on relationships between diversity and transmission, but it’s less clear to me why the results of transmission are better than so-called “snapshot” measures of infection? In this system, transmission occurs from snails to amphibians, and not among amphibian hosts, but this seems quite different from other commonly studied systems, which might involve both inter and intraspecific transmission among hosts.

AUTHOR RESPONSE: The point we are trying to make here is a distinction between pattern and process. The dilution effect generally occurs through a change in the process of transmission (e.g., through encounter reduction). However, given potentially nonlinear, multidirectional relationships between host density, richness, exposure, and infection, simple comparisons between infection levels and host or community metrics may be unlikely to provide reliable insight into the true relationships between these measures. If we were only to measure infection in amphibians, without a consideration of infection pressure, we might fail to detect the very real influences that host richness has on disease. We therefore focus on transmission, or the process of infective parasites moving between hosts, measured as the slope of the relationship between infection pressure (here, density of infective cercariae) and the amount of realized infection in amphibian hosts.

The reviewer raises an interesting point, in that the directionality of transmission is clear in our system, which is different from other systems where directionality is unknown. However, we suggest the need to specifically consider effects on transmission is even greater for those systems; cross-sectional field surveys, which measure the amount of infection at one time point (by prevalence or infected host density) would provide limited information about the processes generating those patterns. This emphasizes the importance of empirically quantifying transmission, either through time or between hosts, to provide clearer insight into directionality of processes, than is available through static patterns of infection alone.

Line 179 – I worry that the authors are conflating transmission in this paragraph (which was modeled as the relationship between pressure and infection risk) with infection risk, which is a term the authors use to describe observed infection (Figure 1). As far as I can tell from the models, there is no statistical evidence that increasing richness broadly reduced individual host infection risk. Increasing richness broadly

reduced transmission (changed the slope of the pressure-risk relationship), but I do not see any results that support the idea of a net negative effect on infection risk. I think that this is really important – because while this study shows strong effects of diversity on transmission, I do not see any results that clearly document a net relationship between diversity and disease, per se.

AUTHOR RESPONSE: We agree, and have worked to revise this section and make our use of terminology more transparent. The Reviewer is correct that the predominant influences of richness were on parasite transmission, or the observed infection load in amphibians while controlling for infection pressure (density of infective stages in the pond). We would contend, however, that this does relate to disease risk in hosts because the number of parasites per host is a direct predictor of pathology. Macroparasites such as trematodes frequently exhibit intensity-dependent pathology in intermediate hosts, which has been well demonstrated for amphibian-trematode interactions. And were host richness lower at a particular site, these results would predict that the success of cercariae in finding and infecting a host would be greater, and hence disease risk would increase.

We now state this more clearly in the revised manuscript:

“First, we evaluated infection success in host individuals, where our response variable was the average number of metacercariae per host. Because infection load per host predicts pathology, this analysis relates directly to disease experienced by individuals.” (see lines 362-365).

*“The resultant infection load (i.e., number of parasites per host) determines host pathology and the transmission opportunities to definitive hosts. Certain trematode species, such as *Ribeiroia ondatrae*, can cause substantial mortality and limb malformations among infected amphibian populations^{28,42,43}.” (see lines 301-304).*

Line 310: Here and elsewhere in the paper, it would be helpful to spell out exactly what value is being interpreted as “parasite transmission”. My understanding is that transmission is the relationship between log10-transformed “infection pressure” and the various responses in the model.

AUTHOR RESPONSE: We have endeavored to be clearer about how we operationally define transmission, including explicit mention in the Introduction and inclusion of a new glossary of terms. In brief, we treat transmission as the amount of infection in amphibian hosts conditional on the amount of exposure. Statistically, this translates into modeling observed infection load in amphibians with infection pressure as an a priori predictor in all models, for which transmission is the slope of this relationship. We then interpret a variable (e.g., density, richness, predators) to alter transmission if we detect a significant interaction between that term and this measure of infection pressure. Our study considers transmission at two biological levels of organization: infection load per host and infection load across the host community.

Line 334 – I do not see these results presented anywhere.

AUTHOR RESPONSE: We elected to remove this allusion to the use of a negative binomial model with or without zero-inflation.

Line 342 – by what were the predictors scaled?

AUTHOR RESPONSE: We now clarify that numeric variables were mean centered (by subtracting the variable mean from each data point) and scaled (by dividing by 1 SD) prior to inclusion in models.

“All numeric predictors were mean-centered and scaled (divided by 1 SD) prior to inclusion, and we ensured that incorporated terms were not collinear (absolute value of $r < 0.7$; correlation matrices of predictor terms are provided in the SI).” (see lines 414-416).

Line 344 – What is the justification for this model-selection approach?

AUTHOR RESPONSE: After additional consideration, we elected to remove the use of likelihood-ratio tests and model reduction as a selection approach. Thus, we now present results of the same, standardized model for each parasite and at both biological levels (i.e., what was previously considered the global model), which includes infection pressure, host density, host richness, predator density, and each of the pairwise interactions with infection pressure. All coefficient estimates are presented in the SI.

Line 363 – Please describe this measurement earlier in the text, where the variable is first defined. It was unclear until now that this value controlled for host density.

AUTHOR RESPONSE: We have added clarifying text where this metric is first introduced. The new text (line 139-140 of the results) reads:

“Total parasite density (quantified as the sum of each host species’ average infection load multiplied by its larval density)...”

Figure 2. How were the values for the black and grey lines (1 spp and 4+ spp) selected? And how were these lines fit? Do these estimates come from the model presented in Table S1? Or do these lines represent some other analysis of the data? I guess the most appropriate way to fit this would be with a post-hoc analysis of the models provided in Table S1 (i.e., a simple-slopes analysis or something similar). Also a bit confused where the coefficient estimates for the inlays come from, since these values are not reported in Table S1 or in the text. Are these from the full model?

AUTHOR RESPONSE: Thank you for these questions. Addressing them has clarified our methods and will hopefully make the figure easier to interpret and understand.

Figure 2 displays model predicted lines at 1 species and 4 species communities, because including lines and error ribbons for all levels of species richness (1-6) would be too busy visually and challenging to interpret. We selected 1 species communities as our ‘low richness’ option because this is simply the lowest richness level observed and is regularly observed in the dataset. While we do observe communities with 5 or 6 amphibian species present, these levels of richness are quite rare (2% of communities), leading to uncertainty in predictions. A community of 4 amphibians is thus both on the high end of observed richness, and well-represented in the dataset (allowing for greater confidence in predictions).

The fitted lines in Figure 2 are the marginal effects predicted from our model using the `ggeffects` package (now noted on lines 648-651). This model is now more clearly specified in lines 409-421 of the Methods, lines 102-112 of the Results, and is available as a Table S1 of the supplement. The `ggpredict` function (in `ggeffects`) generates predictions for infection success (our response variable) across levels of observed infection pressure and at the two levels of host richness (1 or 4 species) while holding all other fixed effects at their mean values, and excluding random effects. The fitted lines are therefore predicted from the global model, and conditional on the other effects in the model. The inlays represent the coefficients from our global model, which are also reported in the Results and in Table S1 of the supplement.

Additionally, we have edited the figure caption to better represent how the lines were fitted.

“Figure 2. Host richness broadly decreases infection success at the individual host scale for four trematode parasites. For each trematode species (labels above panels), the average number of parasites (metacercariae) per chorus frog (Pseudacris regilla) is positively predicted by infection pressure (a proxy for the density of infective cercariae based on the density, average size, and prevalence of infection in snail intermediate hosts). The slope of this relationship is steeper in low richness amphibian communities (1 species; gray line) relative to high richness communities (4 species; black line). Smoothed lines represent marginal effects predicted from ggeffects (i.e., lines denote the effect of host richness on transmission with host and predator densities held at their mean values). Predictor and response variables (infection pressure and infection success, respectively) are $\log_{10} + 1$ transformed. Inlays present coefficients for the interactive effects of host density, predator density, and host richness with infection pressure (asterisks show significant interaction terms with infection pressure). To display all raw datapoints, while maintaining visual alignment with the low and high richness groupings of the fitted lines, raw richness values are binned (gray points are communities of 1 or 2 species, black points are communities of 3 or more species). For all four parasites, the interaction between infection pressure and host richness was negative and significant.” (lines 642-647)

Figure 3. Please clarify where these model coefficients come from. Note that there are no coefficients provided in Table S2, and so I cannot look them up to see what they refer to. My best guess is that this is the Pressure X Richness coefficient estimated separately from each host species (so coming from the three-way interactions)? I find it confusing that the language in this figure does not match up at all with the language in the Results or in Figure 2. Dilution and amplification are not mentioned at all in the results, and it is very difficult to figure out the link between the interaction between infection pressure and species richness and the richness effect on transmission, since there is no operational definition of “transmission”. It is clear from reading the statistical analysis subsection of the Methods section that the effect of infection pressure on infection load is being interpreted as transmission, and so the interaction between this effect and richness is the “dilution” or “amplification” effect, but the link between this figure and the actual text in the Results and Methods needs to be strengthened.

AUTHOR RESPONSE: Yes, the Reviewer is correct that this figure is generated using the coefficients for the pressure-by-richness for each amphibian species. We now include a new table in the supplement that shows these raw model results (rather than only the chi-square results comparing full and reduced models) (Table S3). Additionally, we have altered the language in the figure to make it more precise and enhance alignment with other portions of the manuscript. Rather than showing “amplification” and “dilution”, the top and bottom regions of the figure now display “host richness increases infection success in individuals” and “host richness decreases infection success in individuals”. This aids in interpreting the direction of the effect size and conveys information on the biological scale we are addressing. All terms in these labels are also provided in the new glossary.

Figure 4. Please clarify that these predictions are being generated at mean values for the other variables in these models, rather than at 0.

AUTHOR RESPONSE: We have added text clarifying how predictions were generated in the methods (lines 468-469).

“For the community scale model, we generated predictions at mean values for additional covariates (i.e., host density and predator density).”

Table S1, please specify in the legend how these variables were transformed (i.e., infection pressure is

log-10 transformed), including standardization and mean-centering. Where do these p-values come from? Are they from a likelihood ratio test? Are the full (i.e., not reduced) model results presented anywhere? Do these models also have an estimated intercept? It is also frustrating that parasite names are not consistent with the main text (e.g., Alaria in Figure 2 vs A. marcinae in Table S1).

AUTHOR RESPONSE: We have added information on the transformations and scaling/mean-centering to the table legend. The estimated *P*-values are derived from the summary function for the glmmTMB models. We have now ensured that parasite names and the order in which they are presented are standardized across the figures, tables, and main text.

Reviewers' Comments:

Reviewer #1:

Remarks to the Author:

The authors have taken seriously all of my suggestions and have revised the manuscript accordingly. I also appreciated the conscientious way that they sifted through the comments and suggestions by the other reviewer. The "Response to reviewers" document is a model for how thoughtful, open-minded rejoinders should be put together. I find that the paper is much improved and expect that it will be seen as an unusually thorough, careful, and comprehensive study of diversity-parasite-disease relationships.

Reviewer #2:

Remarks to the Author:

I applaud the authors for the thorough revision and clear response to my previous review. This is a very nice paper!

One point that I still think could use further clarity is how theory relating to the overall number of parasites in the community compares with past studies focusing on the average disease load per individual or prevalence in the host community, which I think reflects the bulk of the published literature at this scale. The paper cited here as justification for a multi-scale approach (Rosenthal et al [citation 17]) measured host prevalence at the community scale, and many other studies across communities (Rottstock et al [Citation 30], and many foundational papers by Charles Mitchell) present community-parasite load, a measure that is relativized by the abundance of hosts in the community. Here, the authors present results that are summed across all hosts in the community, which I believe is quite unique in this field of research. This is not to say that the results are not interesting or relevant, just that a bit more effort might be needed to clarify how these results fit in with other studies at this scale.

Personally, I am not familiar with other papers that study the total number of parasite individuals in the host community, and I could not find any such papers in the references of this manuscript. Although I think this is an interesting scale to explore parasites, it seems quite different from the way the rest of the field might be making comparisons, so I have a bit of concern about whether it is appropriate to use this study to explain scale-dependence from others' work. This comment particularly applies to text at Line 252. In Rosenthal et al, the authors are comparing the percent of hosts infected across the community to the percent of hosts infected for each individual species. Here, the authors are comparing the average intensity of infection in a host to the total number of parasites across all hosts. Numerically, these have very different properties, and so it is not at all surprising that the authors detect different patterns.

I also had some minor concern about the use of the "snapshot" analogy in this paper. I think this language is unnecessarily critical, suggesting that past work was short-sighted and overly limited in scope. What about instead distinguishing the processes of transmission and community assembly from the pattern of infection? To me, this is a much stronger justification for this approach and avoids implicitly criticising past work that I believe remains relevant and important.

Responses to reviewers

REVIEWER COMMENTS

Reviewer #1 (Remarks to the Author):

The authors have taken seriously all of my suggestions and have revised the manuscript accordingly. I also appreciated the conscientious way that they sifted through the comments and suggestions by the other reviewer. The “Response to reviewers” document is a model for how thoughtful, open-minded rejoinders should be put together. I find that the paper is much improved and expect that it will be seen as an unusually thorough, careful, and comprehensive study of diversity-parasite-disease relationships.

AUTHOR RESPONSE: Thank you! We deeply appreciate both the thoughtful suggestions for improving the manuscript, as well as the supportive sentiment throughout the review process.

Reviewer #2 (Remarks to the Author):

I applaud the authors for the thorough revision and clear response to my previous review. This is a very nice paper!

AUTHOR RESPONSE: Many thanks! We appreciate the additional insights and comments, for which we provide more detailed responses below.

One point that I still think could use further clarity is how theory relating to the overall number of parasites in the community compares with past studies focusing on the average disease load per individual or prevalence in the host community, which I think reflects the bulk of the published literature at this scale. The paper cited here as justification for a multi-scale approach (Rosenthal et al [citation 17]) measured host prevalence at the community scale, and many other studies across communities (Rottstock et al [Citation 30], and many foundational papers by Charles Mitchell) present community-parasite load, a measure that is relativized by the abundance of hosts in the community. Here, the authors present results that are summed across all hosts in the community, which I believe is quite unique in this field of research. This is not to say that the results are not interesting or relevant, just that a bit more effort might be needed to clarify how these results fit in with other studies at this scale.

Personally, I am not familiar with other papers that study the total number of parasite individuals in the host community, and I could not find any such papers in the references of this manuscript. Although I think this is an interesting scale to explore parasites, it seems quite different from the way the rest of the field might be making comparisons, so I have a bit of concern about whether it is appropriate to use this study to explain scale-dependence from others’ work. This comment particularly applies to text at Line 252. In Rosenthal et al, the authors are comparing the percent

of hosts infected across the community to the percent of hosts infected for each individual species. Here, the authors are comparing the average intensity of infection in a host to the total number of parasites across all hosts. Numerically, these have very different properties, and so it is not at all surprising that the authors detect different patterns.

AUTHOR RESPONSE: We agree about the importance of drawing a distinction between our metric and that of past work, such as that of Rosenthal and colleagues, and that there is a rarity of diversity-disease studies that have considered total parasite density relative to those focused on prevalence or per-host infection load. While total parasite density has been well investigated in an intra-specific context (e.g., Buck et al. 2017), we are similarly unaware of any field studies evaluating total parasite density summed across host species. We believe this metric has value for understanding the future transmission potential of the parasite population, and because it conveys the parasite's fitness as a function of host diversity, also has evolutionary significance. To make this distinction more transparent, we have removed the text about the Rosenthal study and added in the following new text at Line 228 (including a citation for the Rottstock et al. 2017 experiment):

“Our use of this metric is novel in diversity-disease theory; while prevalence and mean infection load have been common responses measured at both the host and community scale^{17,27}, total parasite infection success is rarely invoked as a response to diversity. Yet, taking the parasite perspective (considering how many parasites successfully infect hosts as a consequence of host diversity) has important implications for the recruitment and future transmission of the parasite population²⁸ and for its potential evolution within the host community.” (see lines 224-230).

I also had some minor concern about the use of the “snapshot” analogy in this paper. I think this language is unnecessarily critical, suggesting that past work was short-sighted and overly limited in scope. What about instead distinguishing the processes of transmission and community assembly from the pattern of infection? To me, this is a much stronger justification for this approach and avoids implicitly criticising past work that I believe remains relevant and important.

AUTHOR RESPONSE: This is a great point. Looking back on the text, it did come across as more critical than intended, and altering this language is a useful improvement to the paper. The point about contrasting more static measures of infection (e.g., snapshots) with efforts to evaluate transmission showed up in three parts of the previous submission. We have modified each of these as follows:

For the first, we altered the sentence by removing the final clause of the sentence about snapshot measures:

“We focused specifically on how diversity affects parasite transmission (Glossary), operationally defined here as the slope of the relationship between the number of trematode infective stages (cercariae) and the number of established parasites in amphibian hosts, rather than relying solely on snapshot measures of host infection levels.” (see lines 81-83).

For the second, we changed the sentence as follows:

“These insights were only made evident by measuring density and community effects on the process of transmission (i.e., parasites’ ability to move among hosts), rather than on ‘snapshot’ measures of infection that do not control for exposure.” (original submission).

“These insights were revealed by measuring density and community effects on the process of transmission (i.e., parasites’ ability to move among hosts), rather than on static measures of infection that do not control for exposure.” (lines 198-201 in revision).

For the final instance, we made the following modifications:

“In contrast, studies that test for correlations between richness and a ‘snapshot’ measure of infection (or disease) are often implicitly assuming that the system is at equilibrium and/or that any feedbacks are instantaneous (no temporal lags).” (original submission).

“In contrast, studies that test for correlations between richness and a ‘snapshot’ measure of infection (or disease) offer insight into the outcome of transmission (i.e., the product of infection pressure and infection success).” (lines 259-261 in revision).

We hope that these changes help to address the reviewer’s comments and would welcome any further suggestions for improvement.